# Knowing What Not to Do: Leverage Language Model Insights for Action Space Pruning in Multi-agent Reinforcement Learning

**Zhihao Liu[1,4]\*, Xianliang Yang[2], Zichuan Liu[3], Yifan Xia[3], Wei Jiang[5], Yuanyu Zhang[6], Lijuan Li[1], Guoliang Fan[1], Lei Song[2], Jiang Bian[2]†**

[1] *The Key Laboratory of Cognition and Decision Intelligence for Complex Systems, Institute of Automation, Chinese Academy of Sciences*
[2] *Microsoft Research Asia*
[3] *Nanjing University*
[4] *University of Chinese Academy of Sciences*
[5] *University of Illinois Urbana-Champaign*
[6] *Guizhou University*

**\*Work done during internship at Microsoft Research Asia.**
**†Corresponding author.**

**Reviewed on OpenReview:** `https://openreview.net/forum?id=T49vPTkIt5`

## Abstract

Multi-agent reinforcement learning (MARL) is employed to develop autonomous agents that can learn to adopt cooperative or competitive strategies within complex environments. However, the linear increase in the number of agents leads to a combinatorial explosion of the action space, which may result in algorithmic instability, difficulty in convergence, or entrapment in local optima. While researchers have designed a variety of effective algorithms to compress the action space, these methods also introduce new challenges, such as the need for manually designed prior knowledge or reliance on the structure of the problem, which diminishes the applicability of these techniques. In this paper, we introduce **E**volutionary action **SPA**ce **R**eduction with **K**nowledge (`eSpark`), an exploration function generation framework driven by large language models (LLMs) to boost exploration and prune unnecessary actions in MARL. Using just a basic prompt that outlines the overall task and setting, `eSpark` is capable of generating exploration functions in a zero-shot manner, identifying and pruning redundant or irrelevant state-action pairs, and then achieving autonomous improvement from policy feedback. In reinforcement learning tasks involving inventory management and traffic light control encompassing a total of 15 scenarios, `eSpark` consistently outperforms the combined MARL algorithm in all scenarios, achieving an average performance gain of 34.4% and 9.9% in the two types of tasks respectively. Additionally, `eSpark` has proven to be capable of managing situations with a large number of agents, securing a 29.7% improvement in scalability challenges that featured over 500 agents. The code can be found in `https://github.com/LiuZhihao2022/eSpark`.

## 1 Introduction

Multi-agent reinforcement learning (MARL) has emerged as a powerful paradigm for solving complex and dynamic problems that involve multiple decision-makers Zhang et al. (2021); Wang et al. (2021). However, the intricacies of agent interplay and the exponential expansion of state and action spaces render the solution of MARL problems difficult. Researchers have proposed the Centralized Training with Decentralized Execution (CTDE) framework Oliehoek et al. (2008) and parameter sharing methods, decomposing the value or policy

functions of a multi-agent system into individual agents and sharing model parameters among all agents. As experimentally verified by many of the most prominent MARL algorithms such as Multi-agent PPO (MAPPO) Yu et al. (2022), QMIX Rashid et al. (2020), QPLEX Wang et al. (2021) or QTRAN Son et al. (2019), these methodologies have been demonstrated to be robust strategies for surmounting the challenges posed by MARL. MARL methods based on parameter sharing and CTDE have achieved notable success in a variety of well-established tasks, including StarCraft Multi-Agent Challenge (SMAC) Li et al. (2023a); Wang et al. (2020), the Multi-Agent Particle Environment (MPE) Lowe et al. (2017), and Simulation of Urban MObility (SUMO) Wei et al. (2019); Lu et al. (2023).

Despite the great success of parameter-sharing CTDE methods, their practicality dwindles in real-world tasks involving large number of agents, such as large-scale traffic signal control Mousavi et al. (2017), wireless communication networks Zocca (2019), and humanitarian assistance and disaster response Meier (2015), where centralized training becomes impractical due to large problem scale Munir et al. (2021). Fully Decentralized Training and Execution (DTDE) methods, such as Independent PPO (IPPO) de Witt et al. (2020), offer a scalable solution where resource consumption does not escalate drastically with an increase in the number of agents. However, due to the lack of consideration for agent interactions, they often struggle to find optimal solutions and fall into local optima. Current strategies for addressing large-scale MARL tasks involve introducing task-specific structures to model agent interactions or dividing agents into smaller, independently trained groups Ying et al. (2023); Chen et al. (2020). These methods, however, are constrained by their dependence on task-related structuring, limiting their applicability to a narrow range of problems.

Additionally, the "curse of dimensionality" presents a significant challenge in multi-agent systems Hao et al. (2022b;a), where agents are required to navigate through an expansive action space saturated with numerous actions that are either irrelevant or markedly suboptimal (relative to states). While humans can deftly employ contextual cues and prior knowledge to sidestep such challenges, MARL algorithms typically engage in the exploration of superfluous and extraneous suboptimal actions Zahavy et al. (2018). Besides, prevailing parameter sharing can exacerbate this exploratory dilemma, as will be elucidated in Proposition 2. The issue occurs primarily because agents with shared parameters often prefer suboptimal policies that present short-term advantages, rather than exploring policies that may potentially deliver higher long-term returns.

Exploration is crucial for overcoming local optima, as it encourages agents to discover potentially better states thus refining their policies. While single-agent exploration techniques like the Upper Confidence Bound (UCB) Auer (2002), entropy regularization Haarnoja et al. (2018), and curiosity-based exploration Groth et al. (2021); Pathak et al. (2017) have shown promising results, they struggle with the escalated complexity in MARL scenarios, compounded by issues like deceptive rewards and the "Noisy-TV" problem Burda et al. (2018). Integrating domain knowledge into exploration could significantly enhance exploration efficiency, by helping identify critical elements and problem structures, thereby aiding in the selection of optimal actions Simon (1956). However, the integration of knowledge into a data-driven framework poses significant challenges, particularly when manual input from domain experts is required, thus reducing its practicality.

Recently, Large Language Models (LLMs) such as GPT-4 Achiam et al. (2023) have shown formidable skills in language comprehension, strategic planning, and logical reasoning across various tasks Yao et al. (2023); Zhu et al. (2023). Although not always directly solving complex, dynamic problems, their inferential and error-learning abilities facilitate progressively better solutions through iterative feedback Ma et al. (2024). The integration of LLMs with MARL presents a promising new avenue by facilitating exploration through the pruning of redundant actions. In this paper, we introduce **E**volutionary action **SPA**ce **R**eduction with **K**nowledge (`eSpark`), a novel approach that utilizes LLMs to improve MARL training via optimized exploration functions, which are used to prune the action space. `eSpark` begins by using LLMs to generate exploration functions from task descriptions and environmental rules in a zero-shot fashion. It then applies evolutionary search within MARL to pinpoint the best performing policy. Finally, by analyzing the feedback on the performance of this policy, `eSpark` reflects and proposes a set of new exploration functions, and iteratively optimizes them according to the aforementioned steps. This process enhances the MARL policy by continuously adapting and refining exploration. To summarize, our contributions are as follows:

1. We introduce the `eSpark` framework, which harnesses the intrinsic prior knowledge and encoding capability of LLMs to design exploration functions for action space pruning, thus guiding the explo-

ration and learning process of MARL algorithms. `eSpark` requires no complex prompt engineering and can be easily combined with MARL algorithms.

2. We validate the performance of `eSpark` across 15 different environments within the inventory management task MABIM Yang et al. (2023) and the traffic signal control task SUMO Behrisch et al. (2011). Combined with IPPO, `eSpark` outperforms IPPO in all scenarios, realizing an average profit increase of 34.4% in the MABIM and improving multiple metrics in SUMO by an average of 9.9%. Even in the face of scalability challenges where the DTDE methods typically encounter limitations, `eSpark` elevates the performance of IPPO by 29.7%.

3. We conduct controlled experiments and ablation studies to analyze the effectiveness of each component within the `eSpark` framework. We first validate the advantages of knowledge-based pruning. Subsequently, we conduct ablation studies to demonstrate that both retention training and LLM pruning techniques contribute to the performance of `eSpark`. These effects are even more pronounced in the more complex MABIM environment. Finally, we confirm the significance of the LLM checker and comprehensive feedback in enhancing code generation and exploration function improvement.

## 2 Related works

**LLMs for code generation.** LLMs have demonstrated remarkable advancements in automated code generation, enabling natural-language-to-code (NL2Code) translation with unprecedented accuracy and efficiency Chen et al. (2021). Recent research has explored various enhancements to LLM-based code generation, including instruction tuning Wei et al. (2021), reinforcement learning with feedback Le et al. (2022), retrieval-augmented approaches Gou et al. (2024), and repository-level code generation Liu et al. (2023). Moreover, specialized code LLMs such as StarCoder Li et al. (2023b), CodeLlama Roziere et al. (2023), and DeepSeek-Coder Zhu et al. (2024) have been developed to improve performance on coding benchmarks like HumanEval and MBPP Chen et al. (2021); Austin et al. (2021).

**LLMs for RL and MARL.** The integration of LLMs into RL and MARL has sparked considerable research interest Sharma et al. (2022); Kwon et al. (2023); Li et al. (2024). Some works Hill et al. (2020); Chan et al. (2019) incorporate the goal descriptions of language models that help in enhancing the generalization capabilities of agents designed to follow instructions. Another work serves LLM as a reward designer for robot control Ma et al. (2024). Further studies have extended this approach to complex tasks involving reasoning and planning Huang et al. (2023; 2022b). Moreover, LLMs have been employed to guide exploration and boost RL efficiency Du et al. (2023); Chang et al. (2023); Hu & Sadigh (2023). However, scaling to high-complexity, real-time, multi-agent settings remains a challenge. Our method mitigates this by generating exploration functions to navigate the policy space, thus facilitating the application to complex MARL scenarios without direct LLM-agent decision-making interaction.

**Action space pruning in RL and MARL.** Pruning the action space has been shown to be effective in guiding agent behaviors in complex environments Lipton et al. (2016); Fulda et al. (2017). Techniques include learning an elimination signal to discard unnecessary actions Zahavy et al. (2018), and employing transfer learning that pre-trains agents to isolate useful experiences for later action refinement Shirali et al. (2023); Lan et al. (2022); Ammanabrolu & Riedl (2018). Some works transform action space pruning into a state-dependent action selection problem, making decisions within the reduced space Sun & Wang (2022); Huang et al. (2022a). Further works use manually designed data structures based on prior knowledge to filter actions Dulac-Arnold et al. (2015); Padullaparthi et al. (2022); Nagarathinam et al. (2020). However, training pruning signals or applying transfer learning is inherently difficult with many agents, and the need for expert knowledge in manual pruning rules hampers their transferability, limiting the applicability of current methods. We harness the abundant knowledge embedded within LLMs for action space pruning, demonstrating universal applicability across a multitude of scenarios.

## 3 Preliminaries

### 3.1 Problem formulation and notations

**Markov game framework.** In our study, we explore a Markov game framework, formally defined by the tuple $\langle N, \mathcal{S}, \mathcal{O}, \mathcal{A}, P, R, \gamma \rangle$. Here $N$ represents the total number of participating agents, $\mathcal{S}$ denotes a well-defined state space, and $\mathcal{O} = \prod_{k=1}^{N} \mathcal{O}_k$ constitutes the combined observation space, $\mathcal{A} = \prod_{k=1}^{N} \mathcal{A}_k$ is the joint action space for all agents involved. The transition dynamics are captured by the probability function $P : \mathcal{S} \times \mathcal{A} \times \mathcal{S} \to [0, 1]$, the reward function $R : \mathcal{S} \times \mathcal{A} \to \mathbb{R}$ maps state-action pairs to real-valued rewards. The discount factor is denoted by $\gamma \in [0, 1]$. All the notations used are summarized in Appendix A.

At each discrete time step $t$, the environment is in state $s^t \in \mathcal{S}$. Each agent $k \in [1, 2, \ldots, N]$ receives an observation $o_k^t \in \mathcal{O}_k$ and draws an action from $a_k^t \sim \pi_k(\cdot \mid o_k^t)$, where $\pi_k : \mathcal{O}_k \times \mathcal{A}_k \to [0, 1]$ denotes the policy of agent $k$, and $\sum_{a_k^t \in \mathcal{A}_k} \pi_k(a_k^t \mid o_k^t) = 1$. The joint actions of all agents $\mathbf{a}^t = (a_1^t, a_2^t, \ldots, a_N^t)$ is drawn from the joint policy $\boldsymbol{\pi}(\cdot \mid s^t) = \prod_{k=1}^{N} \pi_k(\cdot \mid o_k^t)$. Subsequently, a reward $\mathrm{r}^t = R(s^t, \mathbf{a}^t)$ is given based on the current state and joint action. The state transition is determined by $s^{t+1} \sim P(\cdot \mid s^t, \mathbf{a}^t)$.

In this paper, we focus on a fully cooperative scenario where all agents share a common reward signal. The collective objective is to maximize the expected cumulative reward, starting from an initial state distribution $\rho^0$. This collaborative approach emphasizes the alignment of individual agent strategies towards maximizing a unified reward $J(\boldsymbol{\pi})$:

$$J(\boldsymbol{\pi}) = \mathbb{E}_{s^0 \sim \rho^0, \mathbf{a}^{0:\infty} \sim \boldsymbol{\pi}, s^{1:\infty} \sim P} \left[ \sum_{t=0}^{\infty} \gamma^{\mathrm{t}} \mathrm{r}^{\mathrm{t}} \right].$$

**Policy with exploration function.** In the configuration of our study, we introduce the exploration function $E : \mathcal{O}_k \times \mathcal{A}_k \to \{0, 1\}$, indicating whether an action is selectable by agent $k$. For a given policy $\pi_k$ of agent $k$ and an exploration function $E$, we define a new policy $\pi_k^E$ as follows:

$$\pi_k^E(\cdot \mid o_k^t) = \frac{\pi_k(\cdot \mid o_k^t) \cdot E(o_k^t, \cdot)}{\sum_{a_k^t \in \mathcal{A}_k} \pi_k(a_k^t \mid o_k^t) \cdot E(o_k^t, a_k^t)}$$

if $\sum_{a_k^t \in \mathcal{A}_k} \pi_k(a_k^t \mid o_k^t) \cdot E(o_k^t, a_k^t) > 0$; otherwise, $\pi_k^E(\cdot \mid o_k^t) = \pi_k(\cdot \mid o_k^t)$. Consequently, the joint policy for all agents under the guidance of $E$ is defined as:

$$\boldsymbol{\pi}^E(\cdot \mid s^t) = \prod_{k=1}^{N} \pi_k^E(\cdot \mid o_k^t).$$

We define the set of all joint policies as $\{\boldsymbol{\pi}\}$ and the set of all exploration functions as $\{E\}$. Let $\{\boldsymbol{\pi}^E\}$ denote the set of joint policies when subjected to an exploration function $E \in \{E\}$. An exploration function $E$ is non-trivial if it assigns a zero probability to at least one observation-action pair. The following proposition naturally arises from the definition:

**Proposition 1.**

1. *For any $E \in \{E\}$, $\{\boldsymbol{\pi}^E\} \subseteq \{\boldsymbol{\pi}\}$. If $E$ is non-trivial, then $\{\boldsymbol{\pi}^E\} \subset \{\boldsymbol{\pi}\}$.*

2. *For any $\boldsymbol{\pi} \in \{\boldsymbol{\pi}\}$, there exists a non-trivial $E \in \{E\}$ such that $J(\boldsymbol{\pi}^E) \geq J(\boldsymbol{\pi})$.*

An intelligent choice of exploration functions does not diminish our ability to discover optimal policies; instead, it allows us to refine the policy space, thereby enhancing the efficiency of the learning process. The proof of this proposition can be found in Appendix B.

### 3.2 Challenges and motivations

The intricate relationships among multiple agents make it extremely difficult to search for the optimal solution in MARL. Without powerful exploration methods, it is nearly impossible to avoid suboptimal outcomes. We will elaborate on this with an example from the following proposition:

**Proposition 2.** *Let's consider a fully cooperative game with N agents, one state, and the joint action space $\{0,1\}^N$, where the reward is given by $r(\mathbf{0}^0, \mathbf{1}^N) = r_1$ and $r(\mathbf{0}^{N-m}, \mathbf{1}^m) = -mr_2$ for all $m \neq N$, $r_1$, $r_2$ are positive real numbers. We suppose the initial policy is randomly and uniformly initialized, and the policy is optimized in the form of gradient descent. Let p be the probability that the shared policy converges to the best policy, then:*

$$p = 1 - \sqrt[N-1]{\frac{r_2}{r_1 + Nr_2}}.$$

Detailed proof is provided in Appendix B. In the above example, we show that the increase in the number of agents makes it more difficult for MARL algorithms to reach the optimal solution. However, based on the problem context, humans can understand problems from a high-level semantic perspective, and quickly find optimal solutions. As LLMs have demonstrated surprising abilities in semantic understanding, reasoning, and planning across various tasks Yuan et al. (2023); Wang et al. (2023), we conduct a simple experiment to test GPT-4's capability for the issue in Proposition 2, and here is GPT-4's response:

> In a fully cooperative game, all agents work together to maximize the total reward. There are two distinct reward conditions:
>
> 1. When all agents choose action 1, the reward is $r_1$, a positive real number.
>
> 2. When there is any number of agents $m$ (where $0 < m < N$) choosing action 1, the reward is $-mr_2$, where $r_2$ is a positive real number.
>
> All agents should act in a way that avoids the negative reward scenario. The negative reward scenario happens anytime there is a mix of 0's and 1's in the action space, which means some agents are choosing 1 and others are choosing 0. Therefore, the optimal joint action for all agents is to all choose 1.

GPT-4 exhibits reasoning abilities on par with those of humans and directly solves the problem in Proposition 2. Propositions 1 has already shown that an intelligent exploration function can not only reduce the searching space but also improve the final performance. This makes us think about the probability of applying the powerful LLMs to prune the redundant action space and thereby guide the exploration in MARL. In Appendix C, we designed a more complex sequential decision-making example which demonstrates GPT-4 can effectively leverage its reasoning capacity to generate code for pruning redundant action spaces. In the following sections, we propose the `eSpark` framework, which integrates the prior knowledge and inferential capability of LLMs to boost the exploration in MARL.

## 4  Method

In this section, we introduce a novel framework, `eSpark`, which integrates robust prior knowledge encapsulated in LLMs. It improves iteratively through a cycle of trial and error, leveraging the capability of LLMs. Figure 1 illustrates the overall training procedure. `eSpark` is composed of three components: (i) zero-shot generation of exploration functions, (ii) evolutionary search for best performing MARL policy, and (iii) detailed feedback of the policy to improve the generation of exploration functions. We denote the performance of policy $i$ as $G_i$ and the policy feedback as $F_i$ (defined in Section 4.3), and the pseudocode is shown in Algorithm 1.

---

**Algorithm 1 eSpark**

1: **Input:** Initial prompt `prom`, LLM checker $\text{LLM}_c$, LLM code generator $\text{LLM}_g$, the evolution iteration number $N$, and sample batch size $K$
2: **Initialize:** policies $\phi_1^1, \phi_2^1, \ldots, \phi_K^1$
3: **for** $i = 1$ **to** $N$ **do**
4:     // Exploration Function Generation
5:     $E_1, \ldots, E_K \sim \text{LLM}_c(\texttt{prom}, \text{LLM}_g(\texttt{prom}))$
6:     // Retention training
7:     **if** $i \neq 1$ **then**
8:         $\phi_1^i, \phi_2^i, \ldots, \phi_K^i \leftarrow \phi_{\text{best}}^{i-1}$
9:     **end if**
10:     // Evolutionary search
11:     $G_1, F_1 = \phi(E_1), \ldots, G_K, F_K = \phi(E_K)$
12:     // Reflection and Feedback
13:     best $\leftarrow \arg\max_k(G_1, G_2, \ldots, G_K)$
14:     `prom` $\leftarrow$ `prom` : $\text{Reflection}(E_{\text{best}}, F_{\text{best}})$
15: **end for**
16: **Output:** $\phi_{\text{best}}^N$

---

### 4.1 Exploration function generation

LLMs have been demonstrated to possess exceptional capabilities in both code comprehension and generation. To this end, we employ a LLM as **LLM code generator**, denoted as $\text{LLM}_g$, whose task is to understand the objectives of the current environment, and output an exploration function:

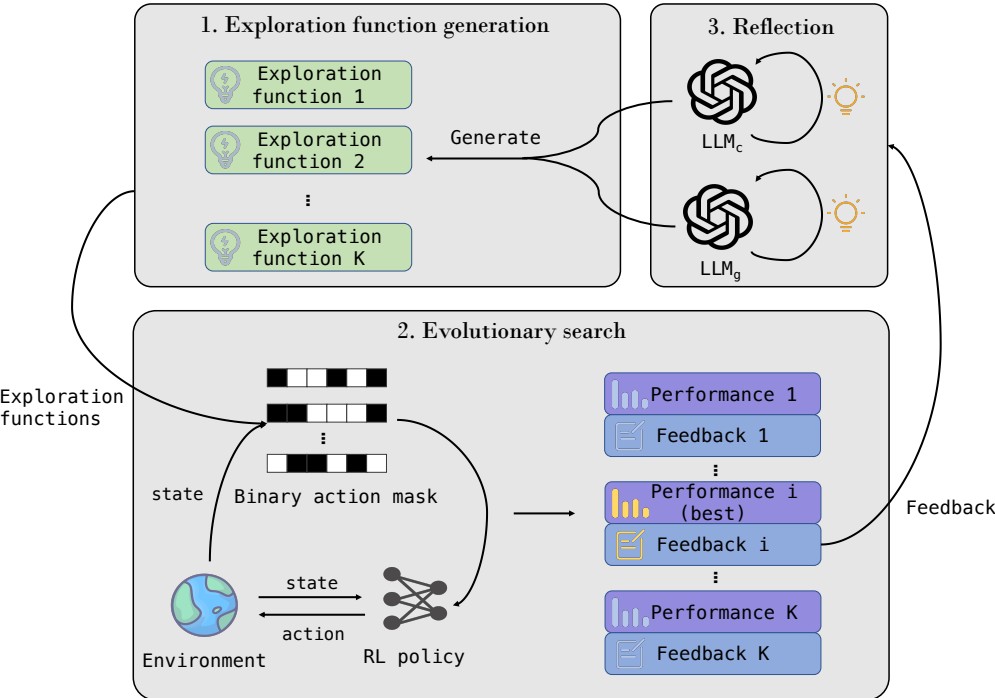

Figure 1: `eSpark` firstly generates $K$ exploration functions via zero-shot creation. Each exploration function is then used to guide an independent policy, and the evolutionary search is performed to find the best-performing policy. Finally, `eSpark` reflects on the feedback from the best performance policy, refines and regenerates the exploration functions for the next iteration.

$$E_1, \ldots, E_K \sim \text{LLM}_g(\texttt{prom}), \tag{1}$$

where `prom` is the prompt for $\text{LLM}_g$, and the generation of $K$ exploration functions is to circumvent the suboptimality that may arise from single-sample generation. The initial `prom` includes an *RL formulation* describing the reward system, state items, transitions, and the action space, alongside a *task description* that specifies the task objectives, expected outputs, and formatting rules. Details on the initial `prom` are provided in Appendix K. We use code for the *RL formulation* as it effectively captures the physical transition dynamics crucial to RL problems, which are always difficult to express precisely through the text alone, especially when environmental complexity increases. Code contexts also improve code generation and clarify environmental semantics and variable roles Ma et al. (2024). In Appendix H, we discuss the impact of different forms of *RL formulation* on the final performance of `eSpark` when the environment code is unavailable.

During the code generation, however, $\text{LLM}_g$ may incorrectly interpret variables and produce logically flawed code. This kind of flawed logic could persist if it is added to the prompt context for the next generation. As research has shown that collaboration among multiple LLMs can enhance the quality and efficacy of the generated contents Chen et al. (2023); Zhang et al. (2023), we introduce the **LLM checker** denoted as $\text{LLM}_c$, which reviews $\text{LLM}_g$'s output to pursue an enhanced generation. $\text{LLM}_c$ uses the same prompt as $\text{LLM}_g$ but is prompted to focus on verifying the accuracy of code relative to environmental transitions and variable specifications. If inconsistencies are found, $\text{LLM}_c$ signals the error, prompting $\text{LLM}_g$ to regenerate the code. The reasons for introducing $\text{LLM}_c$ are further discussed in Section 5.5. Finally, exploration functions are generated by:

$$E_1, \ldots, E_K \sim \texttt{LLM}_c \left( \texttt{prom}, \texttt{LLM}_g(\texttt{prom}) \right). \tag{2}$$

Exploration functions are applied only during the training phase to guide the exploration of MARL. During the execution phase, all exploration functions are removed.

### 4.2 Evolutionary search

During the generation, however, it should be noted that the initially generated exploration function may not always guarantee executability and effectiveness. To address this, eSpark performs an **evolutionary search** paradigm that selects the best-performing policy in one iteration and uses its feedback for subsequent generation Ma et al. (2024). Specifically, eSpark samples a batch of $K$ exploration functions in each generation to ensure there are enough candidates for successful code execution. Performance is assessed at regular checkpoints within an iteration, with the final evaluation based on the average of the last few checkpoints. The policy achieving the highest performance is selected, and the feedback obtained from this policy is integrated to optimize the exploration functions in the following steps.

Due to the dynamic nature of exploration, the exploration function generated based on feedback from the best-performing policy may not be applicable to other policies. As the proof of Proposition 1 demonstrates, when an exploration function is incapable of intelligently pruning, it may even impair the performance of the policy. To this end, we utilize **retention training** to maintain continuity of exploration. Let $\phi_{\text{best}}^{i-1}$ represent best-performing policy from the $(i-1)$-th iteration. For the $i$-th iteration except for the first, at the start of the iteration, we set:

$$\phi_1^i, \phi_2^i, \ldots, \phi_K^i \leftarrow \phi_{\text{best}}^{i-1}. \tag{3}$$

This allows us to match exploration functions with their corresponding policies, subsequently refining performance incrementally. We will verify the impact of retention training in Section 5.4.

### 4.3 Reflection and feedback

Feedback from the environment can significantly enhance the quality of the generated output by LLMs Nascimento et al. (2023); Du et al. (2023). In eSpark, we leverage **policy feedback**, which contains the evaluation of policy performance from various aspects, to enhance the generation of LLMs. This policy feedback may either come from experts or be automatically constructed from the environment, as long as it encompasses insights into the aspects where the current algorithm performs well and areas where it requires improvement. As illustrated in Equation 4, by correlating the best-performing policy feedback $F_{\text{best}}$ and the most effective exploration function $E_{\text{best}}$, LLMs introspect, update the prompt prom, and gear up for the ensuing evolutionary cycle.

$$\texttt{prom} \leftarrow \texttt{prom} : \texttt{Reflection}(E_{\text{best}}, F_{\text{best}}). \tag{4}$$

In our experiments, we generate automated policy feedback from environmental reward signals, as domain experts in relevant fields are not available. We acknowledge that obtaining feedback from human experts can be expensive. Nevertheless, it is important to note that within our framework, the number of rounds for feedback collection is specified by a predefined hyperparameter, which is typically kept low (in our experiments, it is set to 10). Therefore, in scenarios where human experts are accessible, incorporating their insights is feasible and can potentially enhance performance.

## 5 Experiments

### 5.1 Experiment settings

For a comprehensive evaluation of eSpark's capabilities, we perform detailed validations within two distinct industrial environments: the inventory management environment MABIM and the traffic signal control environment SUMO.

- **MABIM setting**: MABIM simulates multi-echelon inventory management by modeling each stock-keeping unit (SKU) as an agent, mirroring real-world operations and profits within the MARL framework. The total reward is composed of multiple reward components. We utilize the total reward to identify the best-performing policy, while those components evaluate the policy's multifaceted performance to generate policy feedback. We focus on three key challenges within inventory management: multiple echelons, capacity constraints and scalability, selecting corresponding scenarios for experiments.

- **SUMO setting**: SUMO is a traffic signal control environment in which each intersection is represented as an agent. It offers a variety of reward functions, and we use "the number of stopped vehicles" as the reward for evolutionary search, while other rewards are for policy feedback. The Average Delay, Average Trip Time, and Average Waiting Time metrics are chosen for evaluation Lu et al. (2023). We employ GESA Jiang et al. (2024) to standardize intersections into 4-arm configurations. Each simulation spans 3600 seconds, with decisions at 15-second intervals.

- **Model setting**: We use IPPO as the base MARL framework for `eSpark` due to its DTDE structure, which is suitable for large-scale challenges. But note that our approach can also be applied as a plugin in other MARL methods. We select GPT-4 for the `LLM`$_c$ and `LLM`$_g$ due to its superior comprehension and generation abilities. The performance of different LLMs can be found in Appendix G. For each scenario, we conduct three runs with a batch size of $K = 16$. `eSpark` has the same number of training steps as the compared MARL baselines, with 10 iterations evenly selected throughout the training process for feedback, reflection, and exploration function editing.

All training jobs are executed with an Intel(R) Xeon(R) Gold 6348 CPU and 4 NVIDIA RTX A6000 GPUs. In Appendix D, we provide a detailed introduction and setting for the environments and model. In Appendix E, we give hyperparameter configurations and descriptions of each baseline method.

## 5.2 Experiment results

In this section, we present the key findings for `eSpark` in the MABIM and SUMO, highlighting the best and second best results in **bold** and underline. More detailed results and computational costs are presented in Appendix F.

### 5.2.1 Performance on MABIM

Figure 2 illustrates the performance of eSpark and the MARL baselines throughout the training process, with the detailed final results presented in Table 1. It can be observed that `eSpark` continuously improves its performance and stabilizes within 10 iterations. With IPPO as the base MARL algorithm, `eSpark` not only outperforms IPPO in all scenarios but also exceeds the performance of all compared baselines in 4 out of 5 scenarios. For an in-depth analysis, we discuss the policy differences between IPPO and `eSpark` in Appendix I, along with `eSpark`'s reflective mechanism and exploration function adjustments in Appendix L. While IPPO struggles to learn the intricate interplay among SKUs, `eSpark` excels particularly in navigating cooperation among SKUs and refining its search in a broad space, leading to marked improvements in managing capacity constraints and multi-echelon coordination.

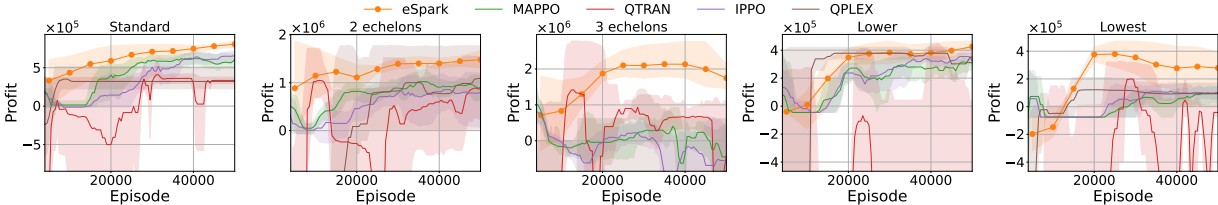

Figure 2: The performance of eSpark and MARL baselines in the MABIM 100 SKUs scenarios. ○ indicates eSpark collects feedback here and regenerates the exploration function. The solid line represents the median, while the shaded region indicates the range between the maximum and minimum values.

Table 1: Performance in MABIM, a higher profit indicates a better performance. The "Standard" scenario features a single echelon with sufficient capacity. The "2/3 echelons" involves challenges of multi-echelon cooperation. The "Lower/Lowest" scenarios are the challenges where SKUs compete for insufficient capacity, while "500 SKUs scenarios" assess scalability. The '-' indicates CTDE algorithms are not researched in the scalability challenges.

| Method | Avg. profits (K) | | | | | | | | | |
| | 100 SKUs scenarios | | | | | 500 SKUs scenarios | | | | |
| | Standard | 2 echelons | 3 echelons | Lower | Lowest | Standard | 2 echelons | 3 echelons | Lower | Lowest |
|---|---|---|---|---|---|---|---|---|---|---|
| IPPO | 690.6 | 1412.5 | 1502.9 | 431.1 | 287.6 | 3021.2 | 7052.0 | 7945.7 | 3535.9 | 2347.4 |
| QTRAN | 529.6 | 1595.3 | 2012.2 | 70.1 | 19.5 | - | - | - | - | - |
| QPLEX | 358.9 | 1580.7 | 704.2 | 379.8 | 259.3 | - | - | - | - | - |
| MAPPO | 719.8 | 1513.8 | 1905.4 | 478.3 | 265.8 | - | - | - | - | - |
| BS static | 563.7 | 1666.6 | 2338.9 | 390.7 | -1757.5 | 3818.5 | 8151.2 | 11926.3 | 3115.1 | -9063.8 |
| BS dynamic | 684.2 | 1554.2 | 2378.2 | **660.6** | -97.1 | 4015.7 | 8399.3 | 11611.1 | **3957.5** | 2008.6 |
| $(S, s)$ | 737.8 | 1660.8 | 1725.2 | 556.9 | 203.7 | 4439.4 | **9952.1** | 10935.7 | 3769.3 | 2212.4 |
| eSpark | **823.7** | **1811.4** | **2598.7** | 579.5 | **405.0** | **4468.6** | 9437.3 | **12134.2** | 3775.7 | **2519.5** |

In Table 1, we also present the performance outcomes of the `eSpark` algorithm in the scaling-up 500 SKUs scenarios. Due to the centralized nature of the CTDE methods, they struggle to scale to large-scale problems and therefore are not presented in the table. Despite IPPO's markedly inferior performance on scenarios when problems scale up, `eSpark` exhibits significant enhancements and consistently achieves optimal results across multiple scenarios. We attribute this improvement to `eSpark`'s action space pruning strategy, which effectively addresses the heightened exploration needs in scenarios with many agents, providing a clear advantage in such complex environments.

### 5.2.2 Performance on SUMO

To further assess `eSpark`'s capabilities across different tasks, we have compiled a summary of results in Table 2 based on the SUMO environment. Similar to the outcomes in MABIM, `eSpark` consistently enhances the performance of the IPPO in all scenarios, and it has outperformed the CTDE baselines as well as domain-specific MARL baselines to achieve the best performance. Notably, even when IPPO alone is capable of good results (as seen in scenarios such as Grid 4×4 and Cologne8), the pruning method designed in `eSpark` does not compromise the effectiveness of IPPO. We will delve further into the analysis of exploration functions produced by `eSpark` in Section 5.3.

### 5.3 `eSpark` learns intelligent pruning methods

Given that `eSpark` employs the prior knowledge of the LLMs to craft its exploration function, our study aimed to investigate two critical aspects: (1) the validity of action space pruning via prior knowledge, and (2) the potential advantages of this method over rule-based heuristic pruning.

To address the questions raised, we devise two pruning strategies. First, we implement a **random pruning** method, wherein agents randomly exclude a portion of actions during decision-making to test the validity of knowledge-based pruning. Secondly, we utilize domain-related OR algorithms to implement **heuristic pruning** methods. For MABIM, actions are pruned using the $(S, s)$ policy and unbound limit, while for SUMO, pruning relies on MaxPressure to keep only a few actions with the highest pressure. The details of these methods are presented in Appendix E.3. Just like `eSpark`, these pruning strategies are integrated with IPPO during training but not execution. We conducted experiments under the same setting in Section 5.1, with results presented in Tables 3 and Table 4.

As shown in the tables, random pruning marginally affects performance by merely altering exploration rates without providing new insights. Heuristic pruning's impact varies with its design and context. In MABIM, $(S, s)$ pruning is less effective in the 100 SKUs scenario, as it restricts the already effective IPPO's exploration in smaller scales. However, it proves beneficial in the 500 SKUs scenario, where it guides the exploration and leads to better results. Upbound pruning consistently underperforms due to its overly simplistic heuristic.

Table 2: Performance in SUMO, including the mean and standard deviation. A lower time indicates a better performance.

| Method | Metric | Grid 4×4 | Arterial 4×4 | Grid 5×5 | Cologne8 | Ingolstadt21 |
|---|---|---|---|---|---|---|
| FTC | Delay | 161.14±3.77 | 1229.68±16.79 | 820.88±24.36 | 85.27±1.21 | 224.96±11.91 |
|  | Trip time | 291.48±4.45 | 676.77±13.70 | 584.54±24.17 | 145.93±0.84 | 352.06±9.29 |
|  | Wait time | 155.66±3.42 | 521.86±13.33 | 441.63±21.13 | 58.92±0.68 | 161.22±7.88 |
| MaxPressure | Delay | 63.39±1.34 | 995.23±77.02 | 242.85±16.04 | 31.63±0.61 | 228.64±15.83 |
|  | Trip time | 174.68±2.05 | 702.09±23.61 | 269.35±9.62 | 95.67±0.62 | 352.30±15.06 |
|  | Wait time | 37.37±1.06 | 511.06±22.55 | 114.36±6.48 | 11.03±0.28 | 159.44±13.34 |
| IPPO | Delay | 48.40±0.45 | 884.73±38.94 | 228.78±11.59 | 27.60±1.70 | 342.97±43.61 |
|  | Trip time | 160.12±0.60 | 506.18±10.39 | 238.03±7.10 | 91.41±1.60 | 464.50±43.30 |
|  | Wait time | 22.69±0.38 | 435.44±77.54 | 91.84±6.31 | 7.70±0.82 | 267.51±40.53 |
| MAPPO | Delay | 51.25±0.58 | 958.43±181.72 | 221.62±20.73 | 32.55±4.66 | 347.59±47.59 |
|  | Trip time | 160.01±0.54 | 757.40±242.00 | 247.56±3.71 | 94.31±1.77 | 480.66±49.46 |
|  | Wait time | 25.41±0.54 | 609.80±255.22 | 97.10±5.22 | 9.39±1.53 | 283.59±43.20 |
| MPLight | Delay | 63.51±0.64 | 1142.98±79.65 | 223.44±16.18 | 37.93±0.45 | **196.74±9.88** |
|  | Trip time | 172.47±0.89 | 583.21±45.84 | 255.49±6.26 | 110.56±1.18 | **331.42±11.79** |
|  | Wait time | 40.32±0.96 | 366.27±58.03 | 126.42±5.31 | 12.98±0.57 | **126.09±13.60** |
| CoLight | Delay | 53.40±1.89 | **820.67±58.65** | 339.66±48.55 | 27.48±1.30 | 296.47±106.82 |
|  | Trip time | 165.77±1.89 | **470.33±12.34** | 305.41±44.43 | 91.66±1.29 | 410.59±97.29 |
|  | Wait time | 27.25±1.64 | **312.47±16.63** | 157.65±35.69 | 9.35±1.09 | 215.98±90.62 |
| eSpark | Delay | **48.36±0.32** | 854.22±68.21 | **209.49±13.98** | 25.39±1.27 | 243.92±15.81 |
|  | Trip time | **159.74±0.44** | 484.87±58.21 | **235.20±6.80** | 89.50±1.36 | 367.57±15.03 |
|  | Wait time | **22.58±0.29** | 328.82±61.70 | **88.38±4.41** | **6.94±0.38** | 180.09±13.84 |

Table 3: Average performance changes on MABIM. All changes are relative to IPPO.

| Method | Avg. profits change ratio (%) | |
|---|---|---|
|  | 100 SKUs | 500 SKUs |
| Random pruning | 2.1 | -0.5 |
| $(S, s)$ pruning | -25.9 | 15.5 |
| Upbound pruning | -23.2 | -32.7 |
| eSpark | **39.1** | **29.7** |

Table 4: Average performance changes on SUMO. All changes are relative to IPPO.

| Method | Avg. time change ratio (%) | | |
|---|---|---|---|
|  | Delay | Trip time | Wait time |
| Random pruning | -0.1 | 2.2 | -2.5 |
| MaxPressure pruning | -0.5 | 1.5 | -0.1 |
| eSpark | **-9.7** | **-5.7** | **-14.3** |

For SUMO, pressure-based pruning does not offer significant benefits. Nevertheless, `eSpark` demonstrates remarkable adaptability across all testing tasks, adeptly selecting pruning methods that substantially enhance results. Its knowledge-based generative technique and evolution capability enable it to master intelligent pruning strategies.

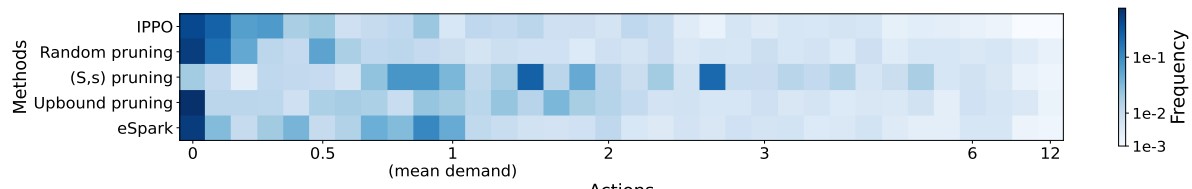

Figure 3: Action selection frequency for IPPO and various pruning methods on the 100 SKUs Lowest scenario. "Actions" represents the replenishment quantity is a multiple of the mean demand within the sliding window. `eSpark` learns not only to minimize restocking but also to diversify with small purchases below the mean demand, balancing demand fulfillment and overflow prevention.

Figure 3 presents a frequency heatmap of action selection for IPPO and various pruning methods in the 100 SKUs Lowest scenario. IPPO learns a minimally restocking strategy, risking unmet demand. Random pruning chooses actions more uniformly yet mirrors IPPO's pattern. $(S, s)$ pruning excessively exceeds mean demand, ignoring no-restock actions and leading to significant overflow. Upbound pruning typically avoids

restocking, but prefers to purchase near the mean demand, which could result in overflow costs. In contrast, `eSpark` adopts a balanced policy, avoiding overstocking while diversifying its minor restocking strategies to meet demand without causing overflow.

### 5.4 `eSpark` benefits from retention training and action space reduction

Extensive research has underscored the importance of reflection in LLM-driven content generation Ma et al. (2024); Nascimento et al. (2023). Herein, we focus on the effects of retention training and action pruning on `eSpark`'s performance.

We first design an ablation experiment, which we refer to as the **eSpark w/o retention**. The model parameters are initialized when an iteration is finished, and the newly generated exploration functions are equipped, after which the training starts from scratch. Given that the initialized model needs a more extensive number of steps to converge, we accordingly triple the training steps per iteration in comparison to the standard `eSpark`. Another ablation retains the retention training, while the only difference is that the LLMs and reflection are removed. We name this experiment **eSpark w/o LLM**. The comparative analysis of these two ablations is delineated in Table 5 and Table 6. The detailed results are shown in Appendix F.3.

Table 5: Average performance change across 100 SKU scenarios in the MABIM environment. All changes are relative to IPPO.

| Method | Avg. profits change ratio (%) |
|---|---|
| eSpark | **39.1** |
| eSpark w/o retention | 24.0 |
| eSpark w/o LLM | -2.8 |

Table 6: Average performance change in the SUMO environment. All changes are relative to IPPO.

| Method | Avg. time change ratio (%) | | |
|---|---|---|---|
| | Delay | Trip time | Wait time |
| eSpark | **-9.7** | **-5.7** | **-14.3** |
| eSpark w/o retention | -9.6 | -4.6 | -11.2 |
| eSpark w/o LLM | -9.1 | -5.0 | -12.8 |

The removal of retention training and LLMs both result in a decline in the performance of the `eSpark`. In the SUMO scenario, the performance gap between the two ablations and the complete `eSpark` is relatively small, whereas it is more pronounced in the MABIM scenarios. This can be attributed to the fact that MABIM involves a greater number of agents and a more complex observation space action space, where a superior pruning can significantly enhance the performance of MARL methods. Additionally, we observe that the lack of LLMs leads to a significant decrease in performance on MABIM, emphasizing the central role of knowledge-based action space pruning within the `eSpark`.

### 5.5 LLM checker and detailed reward feedback promotes the performance of `eSpark`

We go deeper to investigate the impact of feedback prompt design and the LLM checker on the performance of `eSpark`. `eSpark` utilizes detailed reward factors to evaluate policy performance comprehensively. To elaborate on the significance of this design, we perform an ablation experiment, **eSpark w/o r factors**, in which the reward factors are removed, leaving only the total reward.

The introduction of LLM checker comes from the following observations: LLM code generator occasionally produces flawed exploration functions (e.g., variable misuse, misaligned task logic). Although these are usually eliminated during evolutionary search, when the pool of executable exploration functions is small, flawed yet executable functions may still be selected as editing templates, thus hindering further refinement. We conduct the second ablation **eSpark w/o checker** by removing the LLM checker. Both ablations are performed in the MABIM (100 SKUs) and SUMO environments, with the performance averaged in respective environments and presented in Table 7 and Table 8. The detailed results are shown in Appendix F.3.

After removing detailed reward information, `eSpark` showed a significant performance decline in both the MABIM and SUMO environments, particularly in MABIM, where complex transition logic and variable usage exacerbated the impact. Without detailed reward factors, `eSpark` struggled to analyze policy performance and propose targeted improvements, leading to a diminished ability to refine the action space. We provide examples of `eSpark`'s responses with and without reward factors in Appendix J to offer more insights into

Table 7: Average performance change across 100 SKU scenarios in the MABIM environment. All changes are relative to IPPO.

| Method | Avg. profits change ratio (%) |
|---|---|
| eSpark | **39.1** |
| eSpark w/o r factors | 17.8 |
| eSpark w/o checker | 26.2 |

Table 8: Average performance change in the SUMO environment. All changes are relative to IPPO.

| Method | Avg. time change ratio (%) | | |
|---|---|---|---|
| | Delay | Trip time | Wait time |
| eSpark | **-9.7** | **-5.7** | **-14.3** |
| eSpark w/o r factors | -9.4 | -4.8 | -13.5 |
| eSpark w/o checker | -8.1 | -3.1 | -10.8 |

these results. Additionally, the ablation experiment on the LLM checker revealed its critical role in preventing flawed exploration functions from being selected as editing templates, further demonstrating the importance of both detailed reward feedback and the LLM checker in maintaining eSpark's overall performance.

## 6    Conclusions, limitations and future work

We present eSpark, a novel framework for generating exploration functions, leveraging the advanced capabilities of LLMs to integrate prior knowledge, generate code and reflect, thereby refining the exploration in MARL. eSpark has surpassed its base MARL algorithm across all scenarios in both MABIM and SUMO environments. In terms of pruning strategies, pruning based on the prior knowledge from LLMs outshines both random and heuristic approaches. Ablation experiments demonstrate the indispensable role of retention training in accurately improving exploration functions based on policy flaws and enhancing sample efficiency. The LLM checker and detailed policy feedback prompt design together ensure the superior performance of eSpark.

Nevertheless, eSpark also has certain limitations. First, currently eSpark is only applicable to tasks involving homogeneous agents. For heterogeneous agents, a potential method could be to generate distinct exploration functions for each agent; however, this approach becomes impractical when the number of agents is too large. Moreover, eSpark benefits from policy feedback to refine the exploration functions. When feedback is not informative regarding how to modify the exploration (e.g., in tasks with sparse rewards, end-of-episode feedback alone is too limited to develop automated feedback), eSpark may struggle to improve and need extra expert input for effective reflection.

Future work encompasses numerous potential directions. Existing research advocates for assigning different roles or categories to agents Christianos et al. (2021); Wang et al. (2020), which could offer a compromise for the application of eSpark in heterogeneous multi-agent systems. Furthermore, state-specific feedback for more granular improvement represents an intriguing avenue Subramanian et al. (2016). Our future endeavors will investigate these questions, striving to develop algorithms that are robust and exhibit strong generalizability.

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

## A  Notations

All the notations used are summarized in Table 9.

| Notation | Definition |
|:---:|:---:|
| $\boldsymbol{\pi}$ | Combined policy of all agents |
| $\pi_k$ | Policy of agent k |
| $s$ | Ground truth state |
| $\boldsymbol{a}$ | Combined action of all agents |
| $a_k$ | Action of agent k |
| $\boldsymbol{o}$ | Combined observation of all agents |
| $o_k$ | Partial observation of agent k |
| $\gamma$ | Discount factor |
| $r$ | Reward obtained from environment |
| $P(\cdot|s,a)$ | Transition function of environment |
| $E$ | Exploration function generated by LLM |
| $\pi_k^E$ | Policy of agent k under exploration function $E$ |
| $F_k$ | Policy feedback from policy k |
| $G_k$ | Performance evaluation of policy k |

Table 9: Notations and Definitions

## B  Proofs

**Proposition 1.**

*1. For any $E \in \{E\}$, $\{\boldsymbol{\pi}^E\} \subseteq \{\boldsymbol{\pi}\}$. If $E$ is non-trivial, then $\{\boldsymbol{\pi}^E\} \subset \{\boldsymbol{\pi}\}$.*

*2. For any $\boldsymbol{\pi} \in \{\boldsymbol{\pi}\}$, there exists a non-trivial $E \in \{E\}$ such that $J(\boldsymbol{\pi}^E) \geq J(\boldsymbol{\pi})$.*

*Proof.* We begin by offering the proof of the first statement in Proposition 1. We denote $\{\pi_k\}$ as the set of all possible policies for agent $k$, with each $\pi_k$ satisfying the following two conditions:

$$\begin{cases} \pi_k : \mathcal{O}_k \times \mathcal{A}_k \to [0,1] \\ \sum_{a_k \in \mathcal{A}_k} \pi_k(a_k \mid o_k) = 1 \end{cases} \tag{5}$$

Let $\{\pi_k^E\}$ be the set of policies under an exploration function $E$. For every element in $\pi_k^E \in \{\pi_k^E\}$, one of the two situations exists:

1. If $E$ is trivial, then $\pi_k^E(\cdot \mid o_k) = \pi_k(\cdot \mid o_k)$, hence $\pi_k^E \in \{\pi_k\}$.

2. If $E$ is non-trivial, then $\pi_k^E(\cdot \mid o_k) = \frac{\pi_k(\cdot|o_k) \cdot E(o_k,\cdot)}{\sum_{a_k \in \mathcal{A}_k} \pi_k(a_k|o_k) \cdot E(o_k,a_k)}$. It is clear that $0 \leq \pi_k^E(\cdot \mid o_k) \leq 1$ and $\sum_{a_k \in \mathcal{A}_k} \pi_k^E(a_k \mid o_k) = 1$, thus $\pi_k^E \in \{\pi_k\}$.

Therefore, we know that:

$$\{\pi_k^E\} \subseteq \{\pi_k\}. \tag{6}$$

Given that for $\forall \boldsymbol{\pi}(\cdot \mid s^t) = \prod_{k=1}^N \pi_k(\cdot \mid o_k)$, we have $\boldsymbol{\pi} \in \{\boldsymbol{\pi}\}$, where each $\pi_k$ belongs to $\{\pi_k\}$. According to Formula 6, it is known that for $\forall \boldsymbol{\pi}^E(\cdot \mid s^t) = \prod_{k=1}^N \pi_k^E(\cdot \mid o_k)$, it belongs to $\{\boldsymbol{\pi}\}$. Then:

$$\{\boldsymbol{\pi}^E\} \subseteq \{\boldsymbol{\pi}\}. \tag{7}$$

When $E$ is non-trivial, $\pi_k^E \in \{\pi_k\}$ still holds, but $\pi_k \in \{\pi_k^E\}$ may not be true (i.e., when $\pi_k(a_k \mid o_k) > 0$ for $\forall a_k \in \mathcal{A}$, $\pi_k \notin \{\pi_k^E\}$). Hence we can get:

$$\{\pi_k^E\} \subset \{\pi_k\}. \tag{8}$$

In a similar manner, we can deduce that if there exists a $k \in [1, 2, \ldots, N]$ such that $\pi_k \notin \{\pi_k^E\}$, then $\boldsymbol{\pi} \notin \{\boldsymbol{\pi}^E\}$, which means:

$$\{\boldsymbol{\pi}^E\} \subset \{\boldsymbol{\pi}\}. \tag{9}$$

Therefore, we finish the proof of the first statement.

To proof the second statement, it is necessary to introduce a series of variables. We define the value function and state-action function for $\boldsymbol{\pi}$ as follows: $V_{\boldsymbol{\pi}}(s) = \mathbb{E}_{\mathbf{a}^{0:\infty} \sim \boldsymbol{\pi}, s^{1:\infty} \sim P} \left[ \sum_{t=0}^{\infty} \gamma^t \mathbf{r}^t \mid \mathbf{s}^0 = \mathbf{s} \right]$ and $Q_{\boldsymbol{\pi}}(s, a) = \mathbb{E}_{\mathbf{a}^{1:\infty} \sim \boldsymbol{\pi}, s^{1:\infty} \sim P} \left[ \sum_{t=0}^{\infty} \gamma^t \mathbf{r}^t \mid \mathbf{s}^0 = \mathbf{s}, \mathbf{a}^0 = \mathbf{a} \right]$. The advantage function is defined as $A_{\boldsymbol{\pi}}(s, \mathbf{a}) = Q_{\boldsymbol{\pi}}(s, \mathbf{a}) - V_{\boldsymbol{\pi}}(s)$. The joint exploration function is introduced as $\mathbf{E}(s, \cdot) = \prod_{k=1}^{N} E(o_k, \cdot)$. The relationship between $V_{\boldsymbol{\pi}}(s)$ and $Q_{\boldsymbol{\pi}}(s, \mathbf{a})$ can be formulated as:

$$V_{\boldsymbol{\pi}}(s) = \sum_{\mathbf{a} \in \mathcal{A}_k} \boldsymbol{\pi}(\mathbf{a} \mid s) Q_{\boldsymbol{\pi}}(s, \mathbf{a}) \tag{10}$$

For a non-trivial $E$, the value function of $\boldsymbol{\pi}^E$ can be written as:

$$\begin{aligned}
V_{\boldsymbol{\pi}^E}(s) &= \sum_{\mathbf{a} \in \mathcal{A}_k} \frac{\mathbf{E}(s, \mathbf{a}) \boldsymbol{\pi}(\mathbf{a} \mid s)}{\sum_{\mathbf{a} \in \mathcal{A}_k} \mathbf{E}(s, \mathbf{a}) \boldsymbol{\pi}(\mathbf{a} \mid s)} Q_{\boldsymbol{\pi}}(s, \mathbf{a}) \\
&= \frac{1}{\sum_{\mathbf{a} \in \mathcal{A}_k} \mathbf{E}(s, \mathbf{a}) \boldsymbol{\pi}(\mathbf{a} \mid s)} \sum_{\mathbf{a} \in \mathcal{A}_k} \mathbf{E}(s, \mathbf{a}) \boldsymbol{\pi}(\mathbf{a} \mid s) Q_{\boldsymbol{\pi}}(s, \mathbf{a}) \\
&= \frac{1}{\sum_{\mathbf{a} \in \mathcal{A}_k} \mathbf{E}(s, \mathbf{a}) \boldsymbol{\pi}(\mathbf{a} \mid s)} \left[ V_{\boldsymbol{\pi}}(s) - \sum_{\mathbf{a} \in \mathcal{A}_k} (1 - \mathbf{E}(s, \mathbf{a})) \boldsymbol{\pi}(\mathbf{a} \mid s) Q_{\boldsymbol{\pi}}(s, \mathbf{a}) \right].
\end{aligned} \tag{11}$$

Thus, we have:

$$\begin{aligned}
V_{\boldsymbol{\pi}^E}(s) - V_{\boldsymbol{\pi}}(s) &= \frac{V_{\boldsymbol{\pi}}(s) - \sum_{\mathbf{a} \in \mathcal{A}_k} (1 - \mathbf{E}(s, \mathbf{a})) \boldsymbol{\pi}(\mathbf{a} \mid s) Q_{\boldsymbol{\pi}}(s, \mathbf{a})}{\sum_{\mathbf{a} \in \mathcal{A}_k} \mathbf{E}(s, \mathbf{a}) \boldsymbol{\pi}(\mathbf{a} \mid s)} - V_{\boldsymbol{\pi}}(s) \\
&= \frac{\left( 1 - \sum_{\mathbf{a} \in \mathcal{A}_k} \mathbf{E}(s, \mathbf{a}) \boldsymbol{\pi}(\mathbf{a} \mid s) \right) V_{\boldsymbol{\pi}}(s) - \sum_{\mathbf{a} \in \mathcal{A}_k} (1 - \mathbf{E}(s, \mathbf{a})) \boldsymbol{\pi}(\mathbf{a} \mid s) Q_{\boldsymbol{\pi}}(s, \mathbf{a})}{\sum_{\mathbf{a} \in \mathcal{A}_k} \mathbf{E}(s, \mathbf{a}) \boldsymbol{\pi}(\mathbf{a} \mid s)} \\
&= \frac{\sum_{\mathbf{a} \in \mathcal{A}_k} (1 - \mathbf{E}(s, \mathbf{a})) \boldsymbol{\pi}(\mathbf{a} \mid s) V_{\boldsymbol{\pi}}(s) - \sum_{\mathbf{a} \in \mathcal{A}_k} (1 - \mathbf{E}(s, \mathbf{a})) \boldsymbol{\pi}(\mathbf{a} \mid s) Q_{\boldsymbol{\pi}}(s, \mathbf{a})}{\sum_{\mathbf{a} \in \mathcal{A}_k} \mathbf{E}(s, \mathbf{a}) \boldsymbol{\pi}(\mathbf{a} \mid s)} \\
&= -\frac{\sum_{\mathbf{a} \in \mathcal{A}_k} (1 - \mathbf{E}(s, \mathbf{a})) \boldsymbol{\pi}(\mathbf{a} \mid s) A_{\boldsymbol{\pi}}(s, \mathbf{a})}{\sum_{\mathbf{a} \in \mathcal{A}_k} \mathbf{E}(s, \mathbf{a}) \boldsymbol{\pi}(\mathbf{a} \mid s)}.
\end{aligned} \tag{12}$$

When $-\sum_{\mathbf{a} \in \mathcal{A}_k} (1 - \mathbf{E}(s, \mathbf{a})) \boldsymbol{\pi}(\mathbf{a} \mid s) A_{\boldsymbol{\pi}}(s, \mathbf{a}) \geq 0$, which means the expectation of the advantage value for the pruned actions is less than or equal to 0, then $V_{\boldsymbol{\pi}^E}(s) \geq V_{\boldsymbol{\pi}}(s)$. Because for every $s \in \mathcal{S}$, $\sum_{\mathbf{a} \in \mathcal{A}_k} A_{\boldsymbol{\pi}}(s, \mathbf{a}) = \sum_{\mathbf{a} \in \mathcal{A}_k} Q_{\boldsymbol{\pi}}(s, \mathbf{a}) - V_{\boldsymbol{\pi}}(s) = 0$, there always exist actions for which the advantage function values are less than or equal to zero.

As $J(\boldsymbol{\pi}^E) - J(\boldsymbol{\pi}) = \mathbb{E}_{s^0 \sim \rho^0} \left[ V_{\boldsymbol{\pi}^E}(s^0) - V_{\boldsymbol{\pi}}(s^0) \right]$, if an exploration function $E$ can satisfy the condition that for all states $s \in \mathcal{S}$, the inequality $-\sum_{\mathbf{a} \in \mathcal{A}_k} (1 - \mathbf{E}(s, \mathbf{a})) \boldsymbol{\pi}(\mathbf{a} \mid s) A_{\boldsymbol{\pi}}(s, \mathbf{a}) \geq 0$ holds, then it can be guaranteed that $J(\boldsymbol{\pi}^E) \geq J(\boldsymbol{\pi})$.

Therefore, we finish the proof of the second statement. $\qquad \square$

**Proposition 2.** *Let's consider a fully cooperative game with N agents, one state, and the joint action space $\{0, 1\}^N$, where the reward is given by $r(\mathbf{0}^0, \mathbf{1}^N) = r_1$ and $r(\mathbf{0}^{N-m}, \mathbf{1}^m) = -mr_2$ for all $m \neq N$, $r_1$, $r_2$ are positive real numbers. We suppose the initial policy is randomly and uniformly initialized, and the policy is*

*optimized in the form of gradient descent. Let $p$ be the probability that the shared policy converges to the best policy, then:*

$$p = 1 - \sqrt[N-1]{\frac{r_2}{r_1 + Nr_2}}. \tag{13}$$

*Proof.* Clearly, the best policy is the deterministic policy with joint action $(\mathbf{0}^0, \mathbf{1}^N)$.

Now, let the shared policy be $(1 - \theta, \theta)$, where $\theta$ is the probability that an agent takes action 1. The expected reward can be written as:

$$
\begin{aligned}
J(\theta) &= \mathbf{Pr}\left(a^{1:N} = (\mathbf{0}^0, \mathbf{1}^N)\right) \cdot r_1 - \sum_{t=0}^{N-1} \mathbf{Pr}\left(a^{1:N} = (\mathbf{0}^{N-t}, \mathbf{1}_t)\right) \cdot t \cdot r_2 \\
&= \theta^N \cdot r_1 - \sum_{t=0}^{N-1} t \cdot C_N^t \theta^t (1-\theta)^{N-t} \cdot r_2 \\
&= \theta^N \cdot r_1 - \sum_{t=0}^{N} t \cdot C_N^t \theta^t (1-\theta)^{N-t} \cdot r_2 + N \cdot \theta^N \cdot r_2,
\end{aligned}
\tag{14}
$$

where $C_N^t$ is a combinatorial number. We need to simplify $\sum_{t=0}^{N} t \cdot C_N^t \theta^t (1-\theta)^{N-t}$ for further analysis. Notice the structural similarity between the results and the binomial theorem:

$$((1-\theta) + \theta)^N = \sum_{t=0}^{N} C_N^t \theta^t (1-\theta)^{N-t}. \tag{15}$$

We take the derivative of $\theta$ on both sides of Formula 15. Because the left side is constant, its derivative is 0. Then:

$$
\begin{aligned}
0 &= \frac{d \sum_{t=0}^{N} C_N^t \theta^t (1-\theta)^{N-t}}{d\theta} \\
&= \sum_{t=0}^{N} C_N^t t \cdot \theta^{t-1} \cdot (1-\theta)^{N-t} + C_N^t (N-t) \cdot (-1) \cdot (1-\theta)^{N-t-1} \cdot \theta^t \\
&= \sum_{t=0}^{N} C_N^t (1-\theta)^{N-t-1} \theta^{t-1} \left((1-\theta)t - (N-t)\theta\right) \\
&= \sum_{t=0}^{N} C_N^t (1-\theta)^{N-t-1} \theta^{t-1} (t - N\theta).
\end{aligned}
\tag{16}
$$

Thus, we have:

$$
\begin{aligned}
N\theta \sum_{t=0}^{N} C_N^t (1-\theta)^{N-t-1} \theta^{t-1} &= \sum_{t=0}^{N} t C_N^t (1-\theta)^{N-t-1} \theta^{t-1} \\
N\theta \sum_{t=0}^{N} C_N^t (1-\theta)^{N-t} \theta^{t} &= \sum_{t=0}^{N} t C_N^t (1-\theta)^{N-t} \theta^{t}.
\end{aligned}
\tag{17}
$$

Notice that the left side of the equation is the expansion form of Formula 15, and the right side of the equation is the desired Formula, we can get:

$$\sum_{t=0}^{N} t C_N^t (1-\theta)^{N-t} \theta^{t} = N\theta. \tag{18}$$

Bring Formula 18 back to Formula 14, we get:

$$J(\theta) = \theta^N \cdot r_1 - N\theta r_2 + N \cdot \theta^N \cdot r_2. \tag{19}$$

In order to maximise $J(\theta)$, we must maximise $\theta^N \cdot (r_1 + Nr_2) - N\theta r_2$. Since the policy optimization usually adopts a gradient manner, we calculate the derivative of Formula 19 with respect to $\theta$ as:

$$\frac{dJ(\theta)}{d\theta} = N\theta^{N-1}(r_1 + Nr_2) - Nr_2. \tag{20}$$

the point $\theta^* = \sqrt[N-1]{\frac{r_2}{r_1 + Nr_2}}$ is the only zero of $\frac{dJ(\theta)}{d\theta}$. When $\theta \le \theta^*$, $\frac{dJ(\theta)}{d\theta} \le 0$; $\theta \ge \theta^*$, $\frac{dJ(\theta)}{d\theta} \ge 0$.

Remember we are trying to maximize $J(\theta)$ through a gradient way, and then the policy improves the parameters in the direction of the gradient. As the initial policy is randomly and uniformly initialized, the $\theta$ is uniformly distributed in the interval [0,1], then the probability that the shared policy converges to the best policy is:

$$p = 1 - \sqrt[N-1]{\frac{r_2}{r_1 + Nr_2}}. \tag{21}$$

Therefore, we finish the proof of Proposition 2. □

## C   An example of GPT performing action space pruning

In this section, we present a challenging sequential decision-making example to show GPT's capability of reasoning and code generation for pruning redundant action spaces.

GPT's input:

Consider a resource allocation game with n agents.  Each agent has a discrete action space $\{0, 1, 2, 3\}$, representing the amount of resources allocated in each step.  The goal is to maximize the team's cumulative utility while satisfying dynamic constraints that evolves over time.

Game Rules:

1.  Utility Function:  The team's cumulative utility is the sum of all agents' contributions, defined as: $U(a_1, a_2, \ldots, a_n) = \sum_{i=1}^{n} \log(1 + a_i)$

2.  Constraints:  At each time step $t$, a global constraint $C_t$ limits the total allocation $\sum_{i=1}^{n} a_i \le C_t$

3.  Penalty for Violations:  If the total allocation exceeds $C_t$, a penalty proportional to the excess is subtracted from the utility:   $P(\text{excess}) = -k \cdot (\text{excess})^2$, where $k$ is a positive constant.

Please write an exploration function that masks invalid bids and bids likely to lead to inefficient budget usage (e.g., overbidding when the reward is low) for each agent.  The function signature is:

```python
def compute_mask(current_allocations, actions_space, constraints):
    return total_mask
```

current_allocations:  Current resource allocation of all agents.
actions_space:  Available actions for each agent.
constraints:  The dynamic constraint $C_t$.

GPT's output:

In this game, the exploration function must account for the following:

1.  Dynamic Constraints:  Predict whether an action will exceed $C_t$ given the current state and other agents' possible actions.

2.  Utility Contribution:  Estimate the marginal utility of each action and prioritize those with higher expected contributions.

3.  Penalty Avoidance:  Mask actions likely to trigger penalties due to constraint violations.

```python
import numpy as np
def compute_mask(current_allocations, actions_space, constraints):
    total_mask = []
    for action in actions_space:
```

```
        predicted_allocation = sum(current_allocations) + action

        # Check if the action violates the constraint
        if predicted_allocation > constraints:
            total_mask.append(0)  # Mask the action
        else:
            # Estimate marginal utility and penalize low contribution
            marginal_utility = np.log(1 + action)
            penalty = -0.1 * (predicted_allocation - constraints) ** 2 if
                predicted_allocation > constraints else 0
            # Mask actions with low marginal utility or high penalties
            if marginal_utility + penalty < 0.5:  # Threshold chosen
                heuristically
                total_mask.append(0)
            else:
                total_mask.append(1)
    return total_mask
```

# D   Detailed settings

## D.1   MABIM details

MABIM is a simulation environment dedicated to leveraging MARL to tackle the challenges inherent in inventory management problems. Within MABIM, each stock SKU at every echelon is represented as an autonomous agent. The decision-making process of each agent reflects the procurement requirements for the specific SKU at its corresponding echelon.

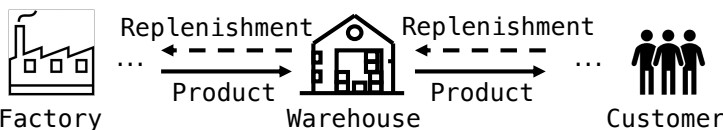

Figure 4: MABIM inventory model.

Each time step involves the agent making decisions regarding replenishment quantities for SKUs and subsequently transitioning the environment to a new state. Let $M \in \mathbb{Z}^+$ be the total warehouses, with the first one being closest to customers, and $N \in \mathbb{Z}^+$ the total SKUs. Given a variable $X \in \{D, S, L \ldots\}$, $X_{i,j}^t$ represents its value for the $j$-th SKU in the $i$-th echelon at step $t$, with $0 \leq i < M$ and $0 \leq j < N$. Given the above notations, the main progression of a step can be described as follows:

$$D_{i+1,j}^{t+1} = R_{i,j}^t \qquad \text{(Replenish)}$$

$$S_{i,j}^t = \min(D_{i,j}^t, I_{i,j}^t) \qquad \text{(Sell)}$$

$$A_{i,j}^t = \sum_{k=0}^{t-1} \mathbb{I}(k + L_{i,j}^k == t) \cdot S_{i+1,j}^t \qquad \text{(Arrive)}$$

$$\gamma_i^t = \min\left(\frac{W_i - \sum_j I_{i,j}^t}{\sum_j A_{i,j}^t}, 1\right), B_{i,j}^t = \lfloor A_{i,j}^t \cdot \gamma_i^t \rfloor \qquad \text{(Receive)}$$

$$I_{i,j}^{t+1} = I_{i,j}^t - S_{i,j}^t + B_{i,j}^t \qquad \text{(Update)}$$

Here, $D, R, S, I, A, B \in \mathbb{Z}^+$ and $\mathbb{I}(\text{condition})$ is an indicator function that returns 1 if the condition is true, and 0 otherwise. For the topmost echelon, orders are channeled to a super vendor capable of fulfilling all

order demands at that level. Orders from other echelons are directed to their immediate upstream echelons, where the demands are satisfied based on the inventory levels of the upper echelons. The demand at the bottom echelon is derived from actual customer orders captured within real-world data sets. The reward function within MABIM is meticulously calibrated based on the economic realities of inventory management, integrating five fundamental elements: sales profit, order cost, holding cost, backlog cost, and excess cost. The summation of these elements constitutes the reward value, thereby incentivizing agents to optimize inventory control for enhanced profitability and operational efficacy.

MABIM incorporates challenges across five key categories: Scaling up, Cooperation, Competition, Generalization, and Robustness. We concentrate on the challenges associated with Scaling up, Cooperation, and Competition, as these challenges not only manifest in inventory management problems but also exist in a broad range of MARL tasks. We catalog the number of agents, challenges and degrees of difficulty within all the experimental scenarios in Table 10. The specific setting of each scenario is given in Table 11:

Table 10: Tasks and corresponding challenges. '+' denotes the extent of the challenges.

| Task name | Agents number | Scaling up | Challenge Cooperation | Competition |
|---|---|---|---|---|
| Standard (100 SKUs) | 100 | | | |
| 2 echelons (100 SKUs) | 200 | | + | |
| 3 echelons (100 SKUs) | 300 | | ++ | |
| Lower capacity(100 SKUs) | 100 | | | + |
| Lowest capacity (100 SKUs) | 100 | | | ++ |
| Standard (500 SKUs) | 500 | + | | |
| 2 echelons (500 SKUs) | 1000 | + | + | |
| 3 echelons (500 SKUs) | 1500 | + | ++ | |
| Lower capacity (500 SKUs) | 500 | + | | + |
| Lowest capacity (500 SKUs) | 500 | + | | ++ |

Table 11: Experiments settings. "#SKU * N" indicates N times the number of SKUs.

| Task name | #Echelon | #SKU | Capacity per echelon |
|---|---|---|---|
| Standard (100 SKUs) | 1 | 100 | #SKU*100 |
| 2 echelons (100 SKUs) | 2 | 100 | #SKU*100 |
| 3 echelons (100 SKUs) | 3 | 100 | #SKU*100 |
| Lower capacity(100 SKUs) | 1 | 100 | #SKU*50 |
| Lowest capacity (100 SKUs) | 1 | 100 | #SKU*25 |
| Standard (500 SKUs) | 1 | 500 | #SKU*100 |
| 2 echelons (500 SKUs) | 2 | 500 | #SKU*100 |
| 3 echelons (500 SKUs) | 3 | 500 | #SKU*100 |
| Lower capacity (500 SKUs) | 1 | 500 | #SKU*50 |
| Lowest capacity (500 SKUs) | 1 | 500 | #SKU*25 |

For each scenario, we carry out three independent runs. Performance is reported as average test set profits from the top model in each run.

## D.2 SUMO details

SUMO is an open-source, highly portable, microscopic and continuous road traffic simulation package designed to handle large road networks. In the SUMO simulation environment, each intersection is conceptualized as an autonomous agent equipped with an array of predefined traffic signal phases. These phases orchestrate

the traffic flow across the intersection's multiple approaches. The selection of these phases, driven by the assessment of live traffic conditions, is aimed at attenuating road congestion and enhancing the fluidity of vehicular movement through the network, thus contributing to the overall efficiency of urban traffic management.

To conduct a thorough evaluation of each algorithm, we select a total of five scenarios from both synthetic and real-world datasets. These datasets encompass a diverse array of intersections, varying in number and type. The intersections are classified according to their configuration into three categories: bi-directional (2-arm), tri-directional (3-arm), and quadri-directional (4-arm), indicating the number of exit points at each junction. We summarize the type of each dataset, the number of intersections included, and the classification of these intersections in Table 12.

Table 12: The categories of each SUMO dataset, along with the number and types of intersections included.

| Dataset | Category | Intersections number | 2-arm | 3-arm | 4-arm |
|---|---|---|---|---|---|
| Grid 4×4 | synthesis | 16 | 0 | 0 | 16 |
| Arterial 4×4 | synthesis | 16 | 0 | 0 | 16 |
| Grid 5×5 Lu et al. (2023) | synthesis | 25 | 0 | 0 | 25 |
| Cologne8 | real-world | 8 | 1 | 3 | 4 |
| Ingolstadt21 | real-world | 21 | 0 | 17 | 4 |

To facilitate a homogeneous observation and action space conducive to the deployment of various MARL algorithms, we employ the GEneral Scenario-Agnostic (GESA) framework to parse each intersection into a standardized 4-arm intersection with eight potential actions.

### D.3 Model details

We employ IPPO as the base MARL algorithm for `eSpark` due to its ability to scale to large-scale MARL challenges. We select GPT-4 as our $\text{LLM}_c$ and $\text{LLM}_g$, specifically opting for the `2023-09-01-preview` version. The temperature of GPT-4 is set to 0.7, with no frequency penalty and presence penalty. For each scenario, we conduct three runs, setting the batch size for each generation of exploration functions to $K = 16$. This batch size is chosen because it guarantees that the initial generation contains at least one executable exploration function for our environment. We limit the number of training iterations to 10, as we observe that the performance for most scenarios tends to converge within this number of iterations.

## E Baseline details

In our experiments, we employed three categories of baselines: OR baselines, MARL baselines and pruning baselines. In Table E, we list the characteristics and environment of all the baselines utilized in our study. In the rest of this section, we will elucidate the underlying principles of each OR baseline, articulate the design of the pruning baselines, and present the hyperparameter for the MARL baselines.

### E.1 OR baselines

### E.1.1 Base stock algorithm

The base stock algorithm constitutes a streamlined and efficacious approach for inventory control, whereby replenishment orders are initiated upon inventory below a predefined threshold level. This policy is traditionally acknowledged as a fundamental benchmark, favored for its straightforwardness and rapid implementation. The computation of the base stock level is facilitated through a programmatic methodology, as explicated in Equation 22:

Table 13: All the baselines used in the experiments

| Algorithm name | OR baseline | MARL baseline CTDE | MARL baseline DTDE | Pruning baseline | Used environment MABIM | Used environment SUMO |
|---|:---:|:---:|:---:|:---:|:---:|:---:|
| Base stock (BS) Arrow et al. (1951) | ✓ | | | | ✓ | |
| $(S, s)$ Blinder (1990) | ✓ | | | | ✓ | |
| FTC Roess et al. (2004) | ✓ | | | | | ✓ |
| MaxPressure Kouvelas et al. (2014) | ✓ | | | | | ✓ |
| IPPO | | | ✓ | | ✓ | ✓ |
| QTRAN | | ✓ | | | ✓ | |
| QPLEX | | ✓ | | | ✓ | |
| MAPPO | | ✓ | | | ✓ | ✓ |
| MPLight Chen et al. (2020) | | | ✓ | | | ✓ |
| CoLight Wei et al. (2019) | | ✓ | | | | ✓ |
| Ramdom pruning | | | | ✓ | ✓ | ✓ |
| $(S, s)$ pruning | | | | ✓ | ✓ | |
| Upbound pruning | | | | ✓ | ✓ | |
| MaxPressure pruning | | | | ✓ | | ✓ |

$$
\begin{aligned}
\textbf{max} \quad & o_{i,j}^t = \bar{p}_{i,j} \cdot S_{i,j}^t - \bar{c}_{i,j} \cdot S_{i+1,j}^t - \bar{h}_{i,j} \cdot I_{i,j}^{t+1} - \bar{c}_{i,j} \cdot T_{i,j}^0 - \bar{c}_{i,j} \cdot I_{i,j}^0 \\
\textbf{s.t} \quad & I_{i,j}^{t+1} = I_{i,j}^t + S_{i+1,j}^{t-\bar{L}_{i,j}} - S_{i,j}^t \\
& T_{i,j}^{t+1} = T_{i,j}^t - S_{i+1,j}^{t-\bar{L}_{i,j}} + S_{i+1,j}^t \\
& S_{i,j}^t = min(I_{i,j}^t, R_{i,j}^t) \\
& T_{i,j}^0 = \sum_{t=-\bar{L}_{i,j}}^{-1} S_{i+1,j}^t \\
& z_{i,j} = I_{i,j}^{t+1} + S_{i+1,j}^t + T_{i,j}^t \\
& z_{i,j} \in \mathbb{R}^+.
\end{aligned}
\tag{22}
$$

In the above equations, $i$, $j$, and $t$ are indexes for the warehouse, SKU, and discrete time, respectively. The indicators $\bar{p}$, $\bar{c}$, $\bar{h}$, and $\bar{L}$ represent the average selling price, cost of procurement, cost of holding, and lead time. The variables $S$, $R$, $I$, and $T$ denote the quantities associated with sales, orders for replenishment, inventory in stock and inventory in transit. $o_{i,j}^t$ describes the profit objective, while $z_{i,j}$ is indicative of the base stock level.

We utilize two approaches for computing stock levels. The first approach, named **BS static**, involves calculating all base stock levels with historical data from the training set, which are then applied consistently to the test set. The levels remain unchanged during the test period. The second approach, termed as **BS dynamic**, computes stock levels directly on the test set relying on historical data and updates on a regular basis.

### E.1.2  $(S, s)$ **algorithm**

The $(S, s)$ inventory policy serves as a robust framework for managing stock levels. Under this policy, a restocking order is triggered once the inventory count falls below a predefined threshold, identified as $s$. The objective of this replenishment is to elevate the stock to its upper limit, designated as $S$. Empirical analyses have substantiated the efficacy of this protocol in optimizing inventory control processes. As a result, it is adopted as a benchmark heuristic, with the aim of algorithmically ascertaining the most efficacious $(S, s)$ parameter for each discrete SKU in the given inventory dataset. In our implementation, we conduct a search

on the training set to identify the optimal values of $s$ and $S$, after which we apply these values consistently to the test set.

### E.1.3 Fixed-time control algorithm

The Fixed-Time Control (FTC) algorithm is a traditional traffic signal control strategy predicated on predefined signal plans. These plans are typically designed based on historical traffic flow patterns and do not adapt to real-time traffic conditions. The FTC operates on a static schedule where the signal phases at intersections change at fixed intervals. This approach is straightforward and easy to implement but may not be optimal under variable traffic conditions due to its lack of responsiveness to dynamic traffic demands.

In our implementation, the FTC follows a fixed sequence of signal phases: 'WT-ET', 'NT-ST', 'WL-EL', 'NL-SL', 'WL-WT', 'EL-ET', 'SL-ST', 'NL-NT'. Here, 'W', 'E', 'N', and 'S' denote westbound, eastbound, northbound, and southbound traffic, respectively, while 'T' indicates through movement, and 'L' signifies a left turn. Each phase has a duration of 30 seconds

### E.1.4 MaxPressure algorithm

The MaxPressure algorithm represents a more advanced traffic signal control strategy that dynamically adjusts signal phases in response to real-time traffic conditions. It calculates the "pressure" at each intersection, defined as the difference between the number of vehicles on the incoming and outgoing lanes. The algorithm aims to optimize traffic flow by selecting signal phases that reduce the maximum pressure across the network, thus alleviating congestion and enhancing network throughput. Unlike FTC, MaxPressure is adaptive and can continuously optimize signal timing based on the current traffic state, making it more suitable for managing fluctuating traffic volumes.

### E.2 Hyperparameters settings for MARL baselines

The following table enumerates the hyperparameters employed during the training process for all MARL baselines. For all test scenarios, training is performed with a uniform suite of hyperparameters that have not undergone specialized fine-tuning.

Table 14: Hyperparameters of MARL Algorithms Used in MABIM and SUMO Environments. '-' indicates that the algorithm is not set or does not contain this hyperparameter.

| Hyperparameter | MABIM environment | | | | SUMO environment | | |
| --- | --- | --- | --- | --- | --- | --- | --- |
| | IPPO | QTRAN | QPLEX | MAPPO | IPPO | CoLight | MPLight |
| Training steps | 5020000 | 5020000 | 5020000 | 5020000 | 2400000 | 2400000 | 2400000 |
| Discount rate | 0.985 | 0.985 | 0.985 | 0.985 | 0.985 | 0.9 | 0.9 |
| Optimizer | Adam | Adam | Adam | Adam | Adam | RMSProp | RMSProp |
| Optimizer alpha | 0.99 | 0.99 | 0.99 | 0.99 | 0.99 | 0.95 | 0.95 |
| Optimizer eps | 1e-5 | 1e-5 | 1e-5 | 1e-5 | 1e-5 | 1e-7 | 1e-7 |
| Learning rate | 5e-4 | 5e-4 | 5e-4 | 5e-4 | 5e-4 | 1e-3 | 1e-3 |
| Grad norm clip | 10 | 10 | 10 | 10 | 10 | - | - |
| Eps clip | 0.2 | - | - | 0.2 | 0.2 | - | - |
| Critic coef | 0.5 | - | - | 0.5 | 0.5 | - | - |
| Entropy coef | 0 | - | - | 0 | 0 | - | - |
| Accumulated episodes | 4 | 8 | 8 | 4 | 4 | 10 | 10 |
| Number of neighbors | - | - | - | - | - | 5 | - |

### E.3 Pruning baselines

### E.3.1 Random pruning

Random pruning is implemented by randomly masking a certain percentage of the available actions. During the action selection process, each agent will have $p$ percent of its available actions randomly masked. To

balance the observability of the pruning's impact with the preservation of the algorithm's capacity to utilize prior experience, we set $p = 0.3$.

### E.3.2  $(S, s)$ pruning

According to the $(S, s)$ algorithm, for a given SKU, a replenishment quantity of $\Delta = S - s$ is ordered when the current inventory level falls below the threshold $s$; otherwise, no order is placed. We extend the reference replenishment quantity $\Delta$ to a range $[r_1 \times \Delta, r_2 \times \Delta]$, where $r_1, r_2$ are both real numbers and $0 \leq r_1 \leq 1$ and $r_2 \geq 1$. Actions within this interval are deemed available, while those outside of this range are masked. In our implementation, we select $r_1 = 0.5$ and $r_2 = 2$.

### E.3.3  MaxPressure pruning

The MaxPressure pruning method utilizes the heuristic concept of "pressure" at an intersection to prune actions. We calculate the pressure associated with each action, and these pressures are then ranked. The actions with the top-k highest pressures are rendered available for selection. Actions not meeting this threshold are subsequently masked.

A standardized intersection warped through the GESA is modeled as a four-arm intersection comprising eight potential actions. We empirically set $k = 4$ to ensure effective pruning while maintaining a sufficient number of available actions.

## F    Additional results

### F.1    Computational costs

We present the token assumption of `eSpark` in Table 15. Since we do not design specific prompts for different scenario tasks within the same environment, we calculate the average token consumption for all scenarios with each environment.

Table 15: Average token assumption for MABIM and SUMO.

| Environment | Token assumption (M) |
|-------------|----------------------|
| MABIM       | 3.0                  |
| SUMO        | 2.6                  |

The training time and GPU memory usage for `eSpark` and the baselines across different scenarios are presented in Table 16 and Table 17.

Table 16: GPU memory usage of eSpark and MARL baselines.

| Method | GPU memory usage (G) | | |
|--------|----------|------------|------------|
|        | Standard | 2 echelons | 3 echelons |
| eSpark | 27.2     | 33.9       | 42.4       |
| IPPO   | 2.3      | 3.4        | 4.4        |
| QTRAN  | 4.2      | 6.5        | 8.7        |

Table 17: Running time of eSpark and MARL baselines.

| Method | Running time (h) | | |
|--------|----------|------------|------------|
|        | Standard | 2 echelons | 3 echelons |
| eSpark | 18       | 25         | 30         |
| IPPO   | 6        | 8          | 12         |
| QTRAN  | 9        | 15         | 21         |

### F.2    Detailed results of the pruning methods

In this section, we provide the detailed results for multiple pruning baselines as discussed in Section 5.3, along with results of `eSpark` for comparison.

Table 18: Detailed performance of various pruning methods in MABIM.

| Method | Avg. profits (K) | | | | | | | | | |
| | 100 SKUs scenarios | | | | | 500 SKUs scenarios | | | | |
| | Standard | 2 echelons | 3 echelons | Lower | Lowest | Standard | 2 echelons | 3 echelons | Lower | Lowest |
|---|---|---|---|---|---|---|---|---|---|---|
| Random pruning | 733.0 | 1407.6 | 1426.6 | 511.6 | 262.0 | 2718.0 | 8667.4 | 9464.1 | 2535.5 | 2202.1 |
| $(S, s)$ pruning | 394.4 | 832.3 | 933.4 | 441.1 | 258.3 | 3884.3 | 9248.3 | 10282.1 | 3517.9 | 2085.6 |
| Upbound pruning | 745.0 | 630.2 | -2.2 | 557.7 | 294.7 | 3261.3 | 2473.6 | 1657.6 | 2833.0 | 2167.7 |
| eSpark | 823.7 | 1811.4 | 2598.7 | 579.5 | 405.0 | 4468.6 | 9437.3 | 12134.2 | 3775.7 | 2519.5 |

Table 19: Detailed performance of various pruning methods in SUMO, includes the mean and standard deviation.

| Method | Metric | Grid 4×4 | Arterial 4×4 | Grid 5×5 | Cologne8 | Ingolstadt21 |
|---|---|---|---|---|---|---|
| Ramdom pruning | Delay | 49.07±0.36 | 858.33±48.20 | 238.57±9.45 | 25.89±1.34 | 353.38±24.39 |
| | Trip time | 160.13±0.58 | 548.08±61.84 | 241.92±9.60 | 89.75±1.26 | 478.53±22.54 |
| | Wait time | 22.66±0.14 | 387.25±43.93 | 93.49±8.09 | 7.03±0.37 | 281.97±21.90 |
| MaxPressure pruning | Delay | 48.78±0.37 | 890.04±121.50 | 234.27±14.28 | 26.26±0.36 | 337.02±62.18 |
| | Trip time | 160.72±0.17 | 533.36±78.08 | 253.68±17.68 | 90.36±0.88 | 448.11±65.32 |
| | Wait time | 23.03±0.56 | 391.96±78.34 | 102.29±10.34 | 7.33±0.12 | 257.98±61.91 |
| eSpark | Delay | 48.36±0.32 | 854.22±68.21 | 209.49±13.98 | 25.39±1.27 | 243.92±15.81 |
| | Trip time | 159.74±0.44 | 484.87±58.21 | 235.20±6.80 | 89.50±1.36 | 367.57±15.03 |
| | Wait time | 22.58±0.29 | 328.82±61.70 | 88.38±4.41 | 6.94±0.38 | 180.09±13.84 |

## F.3 Detailed results of the ablations

In this section, we provide the detailed results for ablations in Table 20 and Table 21, along with results of `eSpark` for comparison.

Table 20: Detailed performance of ablations in MABIM.

| Method | Avg. profits (K) | | | | |
| | Standard | 2 echelons | 3 echelons | Lower | Lowest |
|---|---|---|---|---|---|
| eSpark w/o retention | 719.0 | 1806.1 | 2388.6 | 547.7 | 294.1 |
| eSpark w/o LLM | 754.7 | 1538.9 | 1109.9 | 536.7 | 198.5 |
| eSpark w/o checker | 780.7 | 1741.6 | 2037.6 | 494 | 295.3 |
| eSpark w/o r factors | 758.2 | 1688.1 | 2375.1 | 498.6 | 368.9 |
| eSpark | 823.7 | 1811.4 | 2598.7 | 579.5 | 405.0 |

## G `eSpark`'s performance with different LLMs

We evaluated the performance of `eSpark` under different LLMs. Specifically, we selected the advanced GPT-4o, the relatively less capable GPT-3.5, and the state-of-the-art open-source model DeepSeek-V3 (671B). Experiments were conducted on two inventory management scenarios, and the results are summarized in Table 22. The results indicate that `eSpark` with GPT-4o achieves performance comparable to that of GPT-4. When using GPT-3.5, performance degrades slightly. Although DeepSeek-V3 can effectively support `eSpark`, a performance gap remains compared to GPT-4.

We further assessed compact versions of state-of-the-art open-source LLMs, including DeepSeek-R1-70B and Qwen2.5-72B. Unfortunately, these models performed poorly in task comprehension and failed to generate executable exploration functions. While these compact models show promise, their current capabilities do not yet match those of full-scale models required by eSpark. As shown in Figure 5 as example, the exploration functions generated by Qwen2.5-72B exhibit issues such as variable misuse and logical inconsistencies, highlighting their limited understanding of task requirements.

Table 21: Detailed performance of ablations in SUMO, includes the mean and standard deviation.

| Method | Metric | Grid 4×4 | Arterial 4×4 | Grid 5×5 | Cologne8 | Ingolstadt21 |
|---|---|---|---|---|---|---|
| eSpark w/o retention | Delay | 48.75±0.49 | 851.56±37.98 | 211.07±22.01 | 25.18±0.51 | 246.05±14.88 |
| | Trip time | 160.42±0.54 | 487.05±65.66 | 248.96±15.04 | 89.23±0.49 | 363.28±14.83 |
| | Wait time | 22.97±0.39 | 338.64±67.07 | 97.77±10.25 | 7.14±0.15 | 175.79±12.47 |
| eSpark w/o LLM | Delay | 48.67±0.48 | 854.10±63.38 | 212.25±16.13 | 24.70±0.56 | 257.14±40.50 |
| | Trip time | 159.90±0.56 | 491.49±67.52 | 238.33±9.41 | 88.57±0.60 | 376.46±40.01 |
| | Wait time | 22.99±0.43 | 342.62±73.88 | 90.89±6.60 | 6.69±0.23 | 188.14±37.03 |
| eSpark w/o checker | Delay | 48.29±0.53 | 872.6±94.89 | 215.58±11.29 | 25.22±0.69 | 258.39±14.44 |
| | Trip time | 159.62±0.48 | 511.45±79.67 | 246.56±9.38 | 89.20±0.68 | 383.33±17.07 |
| | Wait time | 22.59±0.48 | 347.43±68.39 | 95.95±6.33 | 6.92±0.28 | 192.96±15.37 |
| eSpark w/o r factors | Delay | 48.54±0.41 | 849.82±44.81 | 216.27±20.01 | 25.29±0.61 | 240.47±19.69 |
| | Trip time | 159.89±0.63 | 479.29±57.12 | 247.28±8.30 | 89.25±0.62 | 373.32±21.19 |
| | Wait time | 22.85±0.36 | 331.69±74.99 | 90.14±6.95 | 6.88±0.19 | 180.90±13.42 |
| eSpark | Delay | 48.36±0.32 | 854.22±68.21 | 209.49±13.98 | 25.39±1.27 | 243.92±15.81 |
| | Trip time | 159.74±0.44 | 484.87±58.21 | 235.20±6.80 | 89.50±1.36 | 367.57±15.03 |
| | Wait time | 22.58±0.29 | 328.82±61.70 | 88.38±4.41 | 6.94±0.38 | 180.09±13.84 |

Table 22: `eSpark`'s performance with different LLMs in MABIM.

| Method | Avg. profits (K) | |
|---|---|---|
| | Standard | Lowest |
| GPT-4 | 823.7 | 405.0 |
| GPT-4o | 829.2 | 397.5 |
| GPT-3.5 | 807.8 | 381.3 |
| DeepSeek-V3 | 778.0 | 370.1 |

```python
def exploration_function(agent_states, supply_chain):
    # Initialize all actions as allowed
    mask = torch.ones((number_of_actions,))
    for sku in agent_states['inventory']:
        current_inventory = agent_states['inventory'][sku]
        in_transit = agent_states['in_transit'][sku]
        lead_time = agent_states['lead_times'][sku]
        # Calculate expected stock after lead time
        future_stock = current_inventory + in_transit[lead_time]

        for action in range(number_of_actions):
            replenishment = action_values[action]  # Assuming action_values are defined

            # Check capacity constraint
            if (future_stock + replenishment) > supply_chain.capacity:
                mask[action] = 0

            # Check demand constraint
            avg_demand = agent_states['history'][sku]['mean']
            if replenishment > 2 * avg_demand:  # Arbitrary threshold
                mask[action] = 0

            # Prevent negative inventory
            if (current_inventory - avg_demand) < 0:
                mask[action] = 0

    return mask, {}
```

Figure 5: Exploration function generated by Qwen2.5-72B. The generated exploration functions exhibit incorrect function signatures and invalid parameter. The function bodies fail to incorporate the required variables, and misuse both the data structures and the expected usage patterns of those variables.

Table 23 presents the token consumption across different LLMs. Token usage is relatively consistent across tasks, enabling users to select an appropriate LLM based on their computational budget and performance requirements.

Table 23: Average token assumption of `eSpark` with different LLMs.

| Environment | Token assumption (M) |
|---|---|
| GPT-4 | 3.0 |
| GPT-4o | 3.1 |
| GPT-3.5 | 2.7 |
| DeepSeek-V3 | 2.8 |

## H `eSpark`'s performance with different RL formulation

When the environment code is unavailable, we consider manually adding descriptions for transitions and rewards. We first introduce the ablation `eSpark w/ lang`, which replaces the environment codes with natural language descriptions. We conduct experiments on selected scenarios in MABIM with 100 SKUs, and the results are shown in Table 24 and Table 25. While the SUMO tasks remain largely unaffected, the MABIM tasks experience noticeable performance declines. This is because the transition logic and variable usage in SUMO are relatively simple, whereas MABIM involves more complex environment transitions and variable interactions. While environment codes provide precise transition dynamics and variable meanings, natural language often lacks this level of detail.

To further explore effective representation for improving performance in complex environments, we also adopt a strategy similar to Text2reward Xie et al. (2023), where experts create a simplified Python-style representation, referred to as `eSpark w/ pyrep`. As shown in Table 24, the kind of representation obtain performance on par with `eSpark`, which indicates that as long as accurate details and sufficient information are included, the language model can effectively understand the environment and `eSpark` can work well.

Table 24: Performance of eSpark and its ablations in the 100 SKUs setting of the MABIM environment.

| Method | Profits (K) | | |
|---|---|---|---|
| | Standard | 3 echelons | Lowest |
| eSpark | 823.7 | 2598.7 | 405.0 |
| eSpark w/ lang | 777.5 | 752.7 | 347.0 |
| eSpark w/ pyrep | 817.4 | 2522.2 | 409.7 |

Table 25: Performance of eSpark and its ablations in SUMO, includes the mean and standard deviation.

| Method | Metric | Time usage (s) | |
|---|---|---|---|
| | | Arterial 4×4 | Ingolstadt21 |
| eSpark | Delay | 851.56±37.98 | 246.05±14.88 |
| | Trip time | 484.84±58.21 | 367.57±15.03 |
| | Wait time | 328.82±61.70 | 180.09±13.84 |
| eSpark w/ lang | Delay | 830.05±75.95 | 243.32±19.43 |
| | Trip time | 495.18±18.60 | 365.65±21.48 |
| | Wait time | 351.32±27.07 | 178.31±19.31 |

## I Policy performance analysis

To gain a deeper understanding of the policy difference between `eSpark` and IPPO, we select the capacity limit and multiple echelons challenges within the 100 SKUs scenario as representative cases, presenting in Figure 6 the daily profit of `eSpark` and IPPO on the test dataset challenged with capacity limit and multiple

echelons. In the capacity limit challenges, a high daily overflow ratio and low fulfillment ratio suggest that IPPO falls short in mastering the adjustment of restocking quantities for individual agents when capacity is limited, leading to overstocking and substantial overflow. Concurrently, this prevents SKUs required by consumers from being accommodated, culminating in an exceedingly low fulfillment ratio. In multiple echelon challenges, the fulfillment ratio at each echelon gradually decreases over time, indicating that IPPO struggles to comprehend and learn the intricate interplay required for cooperation among various echelons, thereby inadequately fulfilling the demands of each echelon. Such shortcomings not only diminish potential profits but also subject the system to considerable backlog expenses. However, through action space pruning, evolutionary search, and reflection, **eSpark** manages to reduce the search within the vast space, selecting the most effective exploration functions and continuously improving. This approach significantly reduces overflow in the capacity limit scenario and successfully learns suitable cooperation methods for multiple echelons.

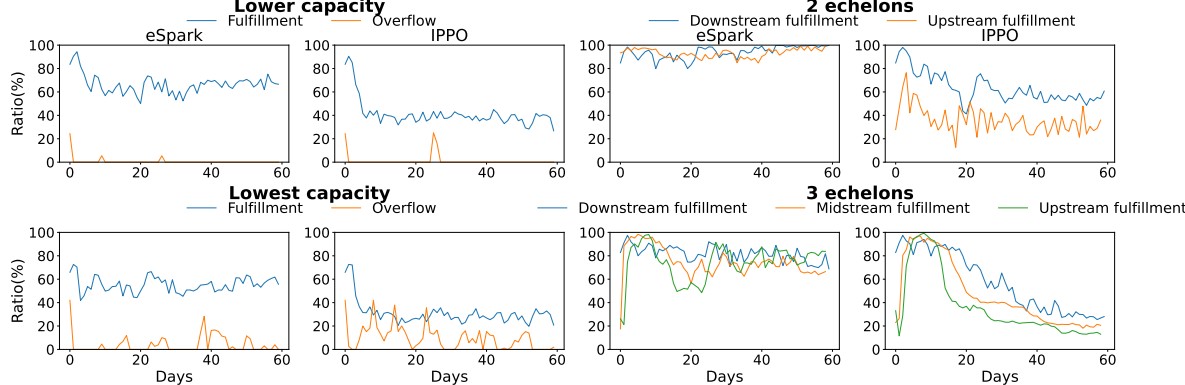

Figure 6: The performance comparison between **eSpark** and IPPO in 100 SKUs scenarios. In capacity-limited scenarios, **eSpark** strives to meet the demands while minimizing overflow costs, boasting a lower overflow ratio and a higher fulfillment ratio. In the multiple-echelon challenge, **eSpark** achieves nuanced collaboration across different echelons, ensuring high fulfillment ratios.

## J  `eSpark`'s response with and without reward factors

We select the policy feedback from one iteration in the Lower scenario of the MABIM environment with 100 SKUs as an example. The complete policy feedback is shown in Figure 7. In the ablation experiment, all reward factors (highlighted in red) are removed. Figure 8 and Figure 9 illustrate the differences in GPT-4's reflections with and without the reward components. After removing the reward feedback, GPT-4 struggles to identify the reasons for poor performance and to thoroughly analyze ways to improve it.

```
We trained a RL policy using the provided exploration function code and tracked the values of the individual
components of the reward function as well as global policy metrics. We also compute the maximum, mean in the early
training stage, mean in the late training stage, mean in all the training stage, minimum values for reward and its
components after every {epoch_freq} epochs. Each element is a one-dimensional array of length n_warehouse,
representing the value of that component on different warehouses:
metric_name: reward, Max: [435592.087], Min: [-1331893.335], Mean in the early training stage: [-212101.5832], Mean
in the late training stage: [338683.3766], Mean in all the training stage: [71512.5884375]
metric_name: profit, Max: [587080.], Min: [430897.], Mean in the early training stage: [493648.], Mean in the late training
stage: [564756.2], Mean in all the training stage: [527295.125]
metric_name: excess_cost, Max: [1648683.5], Min: [17213.5], Mean in the early training stage: [596130.9], Mean in the late
training stage: [123902.4], Mean in all the training stage: [349431.25]
metric_name: order_cost, Max: [60580.], Min: [56490.], Mean in the early training stage: [58864.], Mean in the late training
stage: [58758.], Mean in all the training stage: [58650.625]
metric_name: holding_cost, Max: [456.135], Min: [272.157], Mean in the early training stage: [390.6432], Mean in the late
training stage: [399.6234], Mean in all the training stage: [394.1803125]
metric_name: backlog_cost, Max: [56796.1], Min: [40408.7], Mean in the early training stage: [50364.04], Mean in the late
training stage: [43012.8], Mean in all the training stage: [47306.48125]

Please carefully analyze the policy feedback and provide a new, improved exploration function that can better solve
the task. Some helpful tips for analyzing the policy feedback:
    (1) You can start with "let's think step by step", and then look at each reward individually and think about how
can you improve it
    (2) If the total reward maintains in the same level or even reduce, then you must rewrite the entire exploration
function
    (3) If the values for a certain reward component are near identical throughout, then this means RL is not able
to optimize this component as it is written. You may consider:
        (a) Changing the temperature scale or value of the related mask component so that more action can be
explored
        (b) Re-writing the related mask component
        (c) Discarding the mask component or add a new mask component
```

Figure 7: Policy feedback from last iteration in MABIM 100 SKUs Lower scenario.

```
From the policy feedback, we can identify a few key issues:

1. The total reward is negative in the early training stage and only becomes positive in the later training stages.
This suggests that the agent is not learning effective strategies early on, potentially due to a lack of exploration.
2. The excess cost is quite high, especially in the early training stages. This suggests that the agent is
overstocking the warehouse.
3. The holding cost remains high throughout the training, indicating that the agent is holding onto inventory for
too long.

Given these issues, we need to revise the exploration function to encourage the agent to explore actions that would
result in less overstocking. Here's an improved exploration function:
```

Figure 8: `eSpark`'s reflection with reward factors.

```
From the policy feedback, we can identify a few key issues:

1. The policy feedback indicates that the reward function has a wide range, with the minimum values being
significantly negative and the maximum being positive. The mean reward in the early stage is negative, while it
becomes positive in the late stage. This suggests that the policy improves over time but has considerable variability
in outcomes.
2. The high variance in reward components suggests that the exploration may be too aggressive, leading to
overstocking (incurring high holding costs and overflow costs) or understocking (incurring backlog costs).
3. The exploration function may be too restrictive or lenient, preventing the RL agent from effectively learning the
optimal policy.

We will modify the mask to include a temperature parameter, which will initially be set high to allow for broader
exploration and gradually decrease it as the training progresses. This change will help the agent explore a wider
range of actions early on and focus more on promising actions as it learns.
```

Figure 9: `eSpark`'s reflection without reward factors.

# K   Full prompts

We reference the prompt design outlined in the Eureka Ma et al. (2024) and adapt it specifically for exploration function generation. Our prompt provides general guidance on the design of exploration functions, specific code formatting suggestions, feedback, and recommendations for improvement. We present our prompts for MABIM below.

```
You are an expert in both inventory management learning and reinforcement learning. You will get some exploration
functions, which help mask some actions that are logically unlikely to be selected, to help the exploration in
reinforcement learning tasks as effective as possible.
Your goal is to evaluate whether the given exploration function matches the task description and whether it contains
any illogical errors in the code content, and evaluate whether it's possible to avoid some unreasonable actions,
help the exploration of reinforcement learning. You need to pay special attention to the meaning of each state item
and the logic of the task, making sure to detect
(1) All incorrect use of variables in code
(2) All the parts that don't follow logic.

The exploration function signature can be:
{task_exploration_signature_string}

Your advice can be text or snippets of code, but it should not be the full exploration function code. Most
importantly, remember that your response must begin with either "Code passes check." or "Code fails to pass check.".
Under no circumstance can you begin your answer with other content.
```

Figure 10: System prompt for $\text{LLM}_c$.

```
You are an expert in both inventory management learning and reinforcement learning. You are trying to write some
exploration functions, which helps mask some actions that are logically unlikely to be selected, to help exploration
in reinforcement learning tasks as effective as possible.

Your goal is to write a exploration function for the agent that will mask the actions that's almost impossible to be
chosen in the task described in text.

The exploration function signature can be:
{task_exploration_signature_string}

Your exploration function should only use the variables from the argument list.
Please just give only the exploration function and don't put it in a class. Please make sure that the code is
compatible with numpy (e.g., use numpy array instead of torch tensor).
```

Figure 11: System prompt for $\text{LLM}_g$.

```
Write a exploration function for the following task :
{task_introduction}

The definition of environment and transition are :
{transition_definition}

The agent and state definition is :
{state_definition}

The reward definition is :
{reward_definition}
```

Figure 12: Initial prompt for $\text{LLM}_g$.

```
Please carefully check the exploration function for the following task :
{task_introduction}

The definition of environment and transition are :
{transition_definition}

The agent and state definition is :
{state_definition}

The reward definition is :
{reward_definition}

The requirements of code :
{code_output_tip}
```

Figure 13: Initial prompt for LLM$_c$.

```
Here is the latest exploration function and it's description:
{gpt_response}

Please carefully review this code, check whether it matches the task description and whether it contains any
illogical errors in the code content, and evaluate whether it's possible to improve the exploration and the
performance.
```

Figure 14: LLM$_g$'s feedback to LLM$_c$.

```
We discuss this code with experts, and the code is not approved by experts, and the comments of experts on this code
are as follows:

{checker_feedback}

Please refer to expert advice and combine your own knowledge, fix the problems and provide a new, improved
exploration function!
```

Figure 15: LLM$_c$'s feedback to LLM$_g$.

```
def compute_mask(agent_states: AgentStates, supply_chain: SupplyChain, action_space: list):
    # Here are some code you can refer to when you generate your exploration function.
    ...
    return total_mask, {}
```

Figure 16: Signature of exploration function.

```
Please carefully analyze the policy feedback and provide a new, improved exploration function that can better solve
the task. Some helpful tips for analyzing the policy feedback:
    (1) You can start with "let's think step by step", and then look at each reward individually and think about how
can you improve it
    (2) If the total reward maintains in the same level or even reduce, then you must rewrite the entire exploration
function
    (3) If the values for a certain reward component are near identical throughout, then this means RL is not able
to optimize this component as it is written. You may consider:
        (a) Changing the temperature scale or value of the related mask component so that more action can be
explored
        (b) Re-writing the related mask component
        (c) Discarding the mask component or add a new mask component
```

Figure 17: Output and improvement tips for LLM$_g$.

```
The output of the exploration function should be a total mask (1 denote action is not masked and 0 otherwise). The
code output should be formatted as a python code string: "```python ... ```".

Some helpful tips for writing the exploration function code:
    (1) Your total mask and its component should be a 3-D numpy array in [warehouse_name, sku_type, action_mask].
The masks of the actions that are available are set to 1, otherwise 0. No other elements are allowed to appear in
total mask.
    (2) If you choose to transform a component mask, then you must also introduce a temperature parameter inside the
transformation function; this parameter must be a named variable in the mask function and it must not be an input
variable. Each transformed mask component should have its own temperature variable
    (3) Make sure the type of each input variable is correctly specified; For example, a float input variable should
not be specified as torch.Tensor
    (4) Most importantly, the exploration code's can only use variables defined in its arguments. Under no
circumstance can you introduce new input variables. You only need to give the definition of exploration function and
no other function or class should be defined.
```

Figure 18: Output format for $\text{LLM}_g$.

```
Your output should contain two parts:
    (1) Your response must begin with either "Code passes check." or "Code fails to pass check.". "Code passes
check." means you believe that there are no logical errors in the code and the variables are taken in accordance
with the description of the task. "Code fails to pass check." indicates the provided exploration function contains
logical errors, or you think the code is obviously flawed and you can point out how to facilitate more effective
exploration.
    (2) If you begin with "Code fails to pass check.", you have to explain why the code fails to pass the check and
give your advice on fixing the problems; If you begin with "Code passes check.", you also have to state why each
part is logical and reasonable.

Some common logical errors include:
    (1) Misunderstanding the meaning of the state items, or including syntax errors when using variables
    (2) Illogically handling the state items
    (3) Using multiple unrelated state items to calculate mask component
    (4) The way of combining mask components into total mask is unreasonable
```

Figure 19: Output format for $\text{LLM}_c$.

```
We trained a RL policy using the provided exploration function code and tracked the values of the individual
components of the reward function as well as global policy metrics. We also compute the maximum, mean in the early
training stage, mean in the late training stage, mean in all the training stage, minimum values for reward and its
components after every {epoch_freq} epochs. Each element is a one-dimensional array of length n_warehouse,
representing the value of that component on different warehouses:

        <Reward Feedback Here>
```

Figure 20: Reward feedback and action feedback.

## L   eSpark's exploration function editing

In this section, we demonstrate the reward editing capabilities of eSpark. eSpark is capable of reflecting
upon feedback to optimize the exploration for subsequent iterations.

```
We trained a RL policy using the provided exploration function code and tracked the values of the individual
components of the reward function as well as global policy metrics. We also compute the maximum, mean in the early
training stage, mean in the late training stage, mean in all the training stage, minimum values for reward and its
components after every {epoch_freq} epochs. Each element is a one-dimensional array of length n_warehouse,
representing the value of that component on different warehouses:
metric_name: reward, Max: [435592.087], Min: [-1331893.335], Mean in the early training stage: [-212101.5832], Mean
in the late training stage: [338683.3766], Mean in all the training stage: [71512.5884375]
metric_name: profit, Max: [587080.], Min: [430897.], Mean in the early training stage: [493648.], Mean in the late
training stage: [564756.2], Mean in all the training stage: [527295.125]
metric_name: excess_cost, Max: [1648683.5], Min: [17213.5], Mean in the early training stage: [596130.9], Mean in
the late training stage: [123902.4], Mean in all the training stage: [349431.25]
metric_name: order_cost, Max: [60580.], Min: [56490.], Mean in the early training stage: [58864.], Mean in the late
training stage: [58758.], Mean in all the training stage: [58650.625]
metric_name: holding_cost, Max: [456.135], Min: [272.157], Mean in the early training stage: [390.6432], Mean in the
late training stage: [399.6234], Mean in all the training stage: [394.1803125]
metric_name: backlog_cost, Max: [56796.1], Min: [40408.7], Mean in the early training stage: [50364.04], Mean in the
late training stage: [43012.8], Mean in all the training stage: [47306.48125]

Please carefully analyze the policy feedback and provide a new, improved exploration function that can better solve
the task. Some helpful tips for analyzing the policy feedback:
    (1) You can start with "let's think step by step", and then look at each reward individually and think about how
can you improve it
    (2) If the total reward maintains in the same level or even reduce, then you must rewrite the entire exploration
function
    (3) If the values for a certain reward component are near identical throughout, then this means RL is not able
to optimize this component as it is written. You may consider:
        (a) Changing the temperature scale or value of the related mask component so that more action can be
explored
        (b) Re-writing the related mask component
        (c) Discarding the mask component or add a new mask component
```

Figure 21: Policy feedback from the last iteration.

```
From the policy feedback, we can identify a few key issues:

1. The total reward is negative in the early training stage and only becomes positive in the later training stages.
This suggests that the agent is not learning effective strategies early on, potentially due to a lack of exploration.
2. The excess cost is quite high, especially in the early training stages. This suggests that the agent is
overstocking the warehouse.
3. The holding cost remains high throughout the training, indicating that the agent is holding onto inventory for
too long.

Given these issues, we need to revise the exploration function to encourage the agent to explore actions that would
result in less overstocking. Here's an improved exploration function:
```

Figure 22: **eSpark** reflects upon the policy feedback and proposes the modification.

```python
def compute_mask(agent_states: AgentStates, supply_chain: SupplyChain, action_space: list) -> np.ndarray:
    warehouse_list = supply_chain.get_warehouse_list()
    num_sku = len(agent_states)
    total_mask = np.ones((len(warehouse_list), num_sku, len(action_space)))

    for i, warehouse_name in enumerate(warehouse_list):
        capacity = supply_chain[warehouse_name, "capacity"]
        # Get the volume for all sku
        sku_volume = agent_states[warehouse_name, "volume", "today", "all_skus"]
        # Get the in_stock for all sku
        in_stock = agent_states[warehouse_name, "in_stock", "today", "all_skus"]
        # Get the mean demand for all the skus in the past lookback period
        history_demand_mean = np.average(agent_states[warehouse_name, "demand", "lookback", "all_skus"], axis=1)

        for a, action in enumerate(action_space):
            # Calculate the potential replenishment for all SKUs
            replenishment = history_demand_mean * action
            # If the replenishment will make the total volume of all SKUs exceeds the capacity of the warehouse, mask this action
            if np.sum(sku_volume * (in_stock + replenishment)) > capacity:
                total_mask[i, :, a] = 0

            # Add an additional condition to mask actions that bring in too much replenishment
            if action > 2:
            if action > 1.5:
                total_mask[i, :, a] = 0

                # Add another condition to encourage more exploration in the early training stages
                if agent_states.current_step < agent_states.durations * 0.3 and action < 0.5:
                    total_mask[i, :, a] = 0

    return total_mask, {}
```

Figure 23: Comparison of exploration functions before and after editing.

