# OpenReview forum: "Knowing What Not to Do: Leverage Language Model Insights for Action Space Pruning in Multi-agent Reinforcement Learning"
_TMLR — Accepted by TMLR_

### Review · Reviewer_oyHM · 2025-02-26

**Summary Of Contributions:**

eSpark, a new framework, uses Large Language Models (LLMs) to create exploration functions that enhance Multi-Agent Reinforcement Learning (MARL) algorithms. This integration allows the framework to leverage prior knowledge and improve iteratively through trial and error. The LLM-generated exploration functions are specifically designed for action space pruning, which guides the exploration and learning process of MARL algorithms. Notably, eSpark bypasses the need for complex prompt engineering and seamlessly integrates with existing MARL algorithms.

**Audience:**

Yes

**Claims And Evidence:**

Yes

**Requested Changes:**

See Weakness

**Strengths And Weaknesses:**

Strengths:
• Addresses action space explosion: The paper tackles the problem of combinatorial explosion of the action space in MARL, which leads to algorithmic instability and difficulty in convergence.
• eSpark, in a novel zero-shot approach, utilizes LLMs to generate exploration functions from task descriptions and environmental rules. This innovative method streamlines Multi-Agent Reinforcement Learning (MARL) by pruning unnecessary actions.
• Comprehensive methodology: The eSpark framework includes three components: zero-shot generation of exploration functions, evolutionary search for the best-performing MARL policy, and detailed feedback on the policy's performance to improve the exploration functions.
• Demonstrated effectiveness: eSpark consistently outperforms the combined MARL algorithm in inventory management and traffic light control tasks, achieving an average performance gain of 34.4% and 9.9% respectively. It also shows a 29.7% improvement in scalability challenges with over 500 agents.
• LLM checker and detailed reward feedback: The LLM checker reviews the LLM code generator's output to verify the accuracy of the code. This collaboration enhances the quality and efficacy of the generated content.

Weakness:
• Homogeneous agents: eSpark is currently only applicable to tasks involving homogeneous agents.
• Informative policy feedback: eSpark benefits from policy feedback to refine the exploration functions. When feedback is not informative regarding how to modify the exploration, eSpark may struggle to improve and need extra expert input for effective reflection.

---

### Review · Reviewer_GmtW · 2025-03-05

**Summary Of Contributions:**

This paper studies the use of LLMs to conduct action space pruning, which is implemented through an exploration function generation. In the designed algorithm, LLMs propose, reflect, and improve various exploration functions, such that the learning efficiency and performance of MARL can be improved. Experimental results are comprehensive and supportive to the paper's claims.

**Audience:**

Yes

**Claims And Evidence:**

Yes

**Requested Changes:**

The current organization of the related work section seems a bit unnatural to me.
I would recommend the authors to have a revision such that the LLM for code generation and the LLM for RL/MARL to be better separated. For example, Eureka's paper should be discussed in the section of LLMs for RL / reward design rather than for code generation, in my opinion.

The following work on action space pruning may be worth a cite:

[1] Sun, Hao, and Taiyi Wang. "Toward causal-aware RL: State-wise action-refined temporal difference." arXiv preprint arXiv:2201.00354 (2022).

Discussion on state-action abstraction may further enhance the comprehensiveness of the related work discussion section of this paper --- action abstraction may be considered to be an implicit approach of action pruning.
e.g.,

[2] Huang, Biwei, et al. "Action-sufficient state representation learning for control with structural constraints." International Conference on Machine Learning. PMLR, 2022.


Regarding the method and formalism part, I would suggest using subscripts for the k-th agent all the time. Moreover, a table summarizing definitions of different variables could be helpful for the readers to follow the notation. And this can alleviate the missing definition problem sometimes, e.g., the definition of G in algorithm 1 is missing, and following the conventional definition of G it's reasonable to interpret G as the cumulative return.



Other Questions:

- How challenging would it be in scaling up the proposed method?
- How hard would it be to run the method using local/open-sourced LLMs?
- How sensitive are the results to the LLMs' abilities (this may include both reflection and proposing exploration functions)

**Strengths And Weaknesses:**

The overall idea is clear and well presented in the paper. The authors conducted comprehensive experiments to verify the proposed method, the results look promising.
For potential improvements, please refer to the requested changes section below.

---

> ### Author Response · Authors · 2025-03-18
>
> We thank the reviewer for the valuable suggestions! We are glad that you have read our paper carefully. We hope we can address your concerns below.
>
> **Q1**: The current organization of the related work section seems a bit unnatural to me. I would recommend the authors to have a revision such that the LLM for code generation and the LLM for RL/MARL to be better separated.
>
> **A1**: In the revised version of our paper, we have reorganized the Related Work section to clearly distinguish the applications of LLMs in code generation and RL/MARL tasks, improving readability. We have also added relevant references regarding action-space pruning.
>
> **Q2**: Regarding the method and formalism part, I would suggest using subscripts for the k-th agent all the time. Moreover, a table summarizing definitions of different variables could be helpful for the readers to follow the notation.
>
> **A2**: We have modified our notation to ensure that the k-th agent is always represented with subscript k, maintaining consistency. Besides, we have added the missing definition of $G$ in Algorithm 1 and included a comprehensive table summarizing variable definitions in Appendix A.
>
> **Q3**: How challenging would it be in scaling up the proposed method?
>
> **A3**: eSpark can easily scale to large-scale tasks, primarily due to the following two factors:
>
> 1. **Resource Consumption Control**:
>    - eSpark can be integrated with **DTDE** methods (e.g., IPPO) instead of relying on CTDE/CTCE approaches, avoiding substantial computational overhead in large-scale scenarios.
>    - eSpark **only aggregates feedback from all agents at the end of each iteration** and updates the exploration function accordingly. This prevents the exponential increase in computational cost as the number of agents grows. Thus, even in large-scale tasks, **eSpark maintains stable and manageable token consumption**.
> 2. **Exponential Action-Space Pruning**:
>    - eSpark leverages LLM-generated exploration functions to prune the action space.
>    - In multi-agent environments, the action space grows exponentially with the number of agents. However, eSpark’s exploration function can also prune redundant actions exponentially, significantly improving training efficiency.
>
> These features make eSpark well-suited for large-scale MARL tasks.
>
> **Q4**: How hard would it be to run the method using local/open-sourced LLMs?
>
> **A4**: eSpark can be directly used with local LLMs. Our implementation is designed with a modular architecture, making it straightforward to replace the generation components (i.e., the LLM-based code generator and checker) with a local LLM. Users can seamlessly switch to open-source or private LLMs.
>
> To assess the impact of using local LLMs for eSpark, we conducted experiments using **deepseek-v3 (671b version)** on three inventory management scenarios. The results are shown in the following table:
>
> | LLM Model   |          | Performance |        |
> | ----------- | -------- | ----------- | ------ |
> |             | Standard | Lower       | Lowest |
> | GPT-4       | 823.7    | 579.5       | 405.0  |
> | deepseek-V3 | 778.0    | 506.8       | 370.1  |
>
> The results indicate that deepseek-v3 performs slightly worse than GPT-4 across different scenarios but still supports eSpark effectively.
>
> **Q5**: How sensitive are the results to the LLMs' abilities (this may include both reflection and proposing exploration functions)
>
> **A5**: We evaluated eSpark using different versions of LLMs, including **GPT-3.5, GPT-4o, and deepseek-V3**, as both the code generator and checker. The results are shown below:
>
> | LLM Model   |          | Performance |
> | ----------- | -------- | ----------- |
> |             | Standard | Lowest      |
> | GPT-4       | 823.7    | 405.0       |
> | GPT-4o      | 829.2    | 397.5       |
> | GPT-3.5     | 807.8    | 381.3       |
> | deepseek-V3 | 778.0    | 370.1       |
>
> The results show that **eSpark performs similarly when using GPT-4o compared to GPT-4**. When using **GPT-3.5 or deepseek-V3**, performance slightly declines, but the drop is not substantial.
>
> We also tested the most advanced distilled open-source LLM versions, such as **deepseek-R1-70b** and **qwen2.5-72b**. Unfortunately, this model struggled with task comprehension and failed to generate executable exploration functions. While distilled versions have potential, they have yet to achieve the performance of full-scale models for eSpark.

---

> > ### Comment · Reviewer_GmtW · 2025-04-15
> >
> > I appreciate the authors' efforts in adding new experiments and their detailed responses. I particularly appreciate their new experiments with the open-source DeepSeek-V3 model and the comprehensive results. My initial concerns are well addressed.
> >
> > Further suggestions:
> > - If the authors could also note the cost in the comparison table when comparing different LLMs, it would be more helpful and clearer for readers to understand the cost-performance trade-off.
> > - I really appreciate the authors' disclosure of their failed experiments. If the authors could add those to the paper with some analysis, it would be an additional contribution and meaningful signal to the community.

---

> > > ### Author Response · Authors · 2025-05-08
> > >
> > > We are pleased to hear that our responses addressed your concerns. In Appendix G, we have included the token consumption of eSpark when powered by different LLMs. Overall, the token usage across different LLMs is relatively similar, which allows users to select a suitable model based on their budget constraints.
> > >
> > > Additionally, we have included an example in Appendix G illustrating a failed case where the Qwen2.5-72B model was used to generate exploration functions. The generated code features incorrect function signatures and invalid parameters. Furthermore, the function body fails to include the required variables and misuses both the data structures and the expected usage patterns. This highlights the importance of an LLM’s strong comprehension and code-generation capabilities for eSpark to function effectively.

---

### Review · Reviewer_ZPNz · 2025-03-07

**Summary Of Contributions:**

This paper introduces eSpark, a novel framework that integrates LLMs with Multi-Agent Reinforcement Learning (MARL), using prior knowledge of the LLMs to enhance exploration efficiency via action space pruning. The key contributions of the paper are:
1. Proposing a novel use of LLM to generate code with the aim of pruning redundant action space in MARK environments.
2. The eSpark framework iteratively refines exploration functions using a feedback loop of Self-Improving Exploration via Evolutionary Search.
3. Experimental Evaluation of the method in Multi-Agent Domains of Inventory management and traffic signal control.

**Audience:**

Yes

**Claims And Evidence:**

Yes

**Requested Changes:**

**Questions**
1. While Proposition 1 guarantees that pruning does not eliminate optimal solutions, could excessive pruning still limit exploration if the LLM incorrectly removes useful actions?
2. Did the authors perform comparative analysis with open source LLMs such as Llama or Mistral? I believe it would've highlighted both limitations and feasibility of such widely available models for use with this technique.
3. What happens when the LLM generates flawed or inconsistent rules?

**Recommendations**
- I beileve the paper would benefit from an analysis/experiments of how the LLM usage affects the computational cost when generating useful outcome with this technique. This will allow the readers to gauge the efficacy of this method for their use cases.

The following recent paper on multi-task RL and pruning might be worth citing due to its relevance:
1. Arnob et al. "Efficient Reinforcement Learning by Discovering Neural Pathways", NeurIPS 2024.

**Strengths And Weaknesses:**

**Strengths:**
1. Innovative Use of LLMs for Action Space Pruning.
2. Improved Learning Efficiency and Performance.
3. The paper demonstrates how LLMs can serve as external knowledge sources.
4. Good experimental results.
5. Paper well written and easy to follow. I enjoyed reading it.


**Weaknesses:**
1. Although experimental results are great, they are demonstrated on limited number (two) of tasks.
2. Running multiple LLM queries per iteration may introduce high computational cost and response delays, which are not deeply analyzed in the paper.
3. eSpark relies on a proprietary LLM (GPT-4) to generate exploration functions, which may limit reproducibility.

---

> ### Author Response · Authors · 2025-03-18
>
> Thank you for your detailed feedback and thoughtful questions. We appreciate your insights and hope our responses address your concerns.
>
> **Q1**: While Proposition 1 guarantees that pruning does not eliminate optimal solutions, could excessive pruning still limit exploration if the LLM incorrectly removes useful actions?
>
> **A1**: **Excessive pruning may limit exploration and decrease policy performance**. Proposition 1 demonstrates that for any given policy, there always exists an exploration function $E$ such that the pruned policy outperforms the original policy. However, this theoretical guarantee assumes that **the exploration function is intelligently designed**. If $E$ is not intelligent enough (see the Proposition 1’s second proof result in Appendix B), then pruning may indeed degrade policy performance.
>
> As shown in Table 3 and Table 4, naive pruning methods (such as random pruning or heuristic-based pruning) may fail to enhance performance. In contrast, eSpark leverages LLM's semantic understanding to perform selective pruning, ensuring that it optimizes exploration efficiency while retaining critical actions.
>
> **Q2**: Did the authors perform comparative analysis with open source LLMs such as Llama or Mistral? I believe it would've highlighted both limitations and feasibility of such widely available models for use with this technique.
>
> **A2**: We appreciate the reviewer’s suggestion. We evaluated the performance of the open-source **deepseek-v3 (671B version)** as a code generator and checker in three inventory management scenarios. The experimental results are presented below:
>
> | LLM Model   |          | Performance |        |
> | ----------- | -------- | ----------- | ------ |
> |             | Standard | Lower       | Lowest |
> | GPT-4       | 823.7    | 579.5       | 405.0  |
> | deepseek-V3 | 778.0    | 506.8       | 370.1  |
>
> The results indicate that **although deepseek-V3 can effectively support eSpark, there is a performance gap compared to GPT-4**. In the next version of eSpark, we plan to further evaluate other open-source LLMs, such as **Llama and Mistral**, to provide a comprehensive analysis of eSpark's adaptability across different LLMs.
>
> **Q3**: What happens when the LLM generates flawed or inconsistent rules?
>
> **A3**: We attribute defects in LLM-generated exploration functions to two primary sources:
>
> 1. **Syntax or logical errors**, such as misusing variables or failing to account for task transitions. In such cases, the exploration function may be infeasible.
> 2. **Inadequate mask structures or suboptimal hyperparameter selection**, leading to ineffective guidance for policy learning.
>
> To address the first issue, we introduce a dedicated LLM checker to verify whether each generated exploration function contains syntax or logical errors. This role assignment method has been proven effective in several studies [1,2]. For the second issue, we employ evolutionary search to select the best-performing exploration functions. Furthermore, we incorporate an automatic feedback mechanism for the reward factors within this iteration, enabling eSpark to further refine and improve the shielding functions based on the feedback received.

---

> > ### Author Response · Authors · 2025-03-18
> >
> > **Q4**: The paper would benefit from an analysis/experiments of how the LLM usage affects the computational cost when generating useful outcome with this technique.
> >
> > **A4**: We agree that analyzing the computational complexity and resource consumption of eSpark is essential for assessing its effectiveness across different application scenarios. To this end, we provide a detailed analysis of computational costs, including **runtime cost, LLM token consumption, and memory usage**, to comprehensively evaluate the computational burden of eSpark under tasks of varying scales.
> >
> >
> >
> > **1. Runtime Cost Analysis**
> >
> >
> >
> > eSpark consists of three key components: Exploration Function Generation, Evolutionary Search, and Policy Feedback & Refinement. We analyze their respective computational cost as follows:
> >
> > ​	•	**Exploration Function Generation**: The computational cost of this step is primarily determined by LLM inference time, which is influenced by prompt length, LLM model complexity, and the length of the generated code.
> >
> > ​	•	**Evolutionary Search**: This step involves policy-environment interactions guided by exploration functions. Its computational cost depends on the reinforcement learning algorithm used and the complexity of the exploration function.
> >
> > ​	•	**Policy Feedback and Refinement**: The main computational overhead comes from LLM processing feedback and re-editing the exploration function, similar to the cost of the exploration function generation step.
> >
> > Notably, since the generation, evolutionary search, and refinement of different exploration functions in eSpark are **mutually independent**, we have implemented a **highly parallelized computational framework** in our codebase. This ensures that the overall computational cost does not increase linearly with the number of exploration functions. A detailed runtime comparison can be found in Appendix E.1, Table 17.
> >
> >
> >
> > **2. LLM Token Consumption and Memory Usage**
> >
> > ​	•	We provide a detailed breakdown of LLM token consumption and memory usage across different scenarios in Table 15 and Table 16.
> >
> > ​	•	Despite relying on LLM for exploration function optimization, **eSpark’s token consumption does not scale with the number of agents**. This is because eSpark only aggregates statistical feedback at the end of each iteration,  rather than invoking the LLM for step-by-step policy decisions.
> >
> > Our experimental and computational analyses demonstrate that eSpark exhibits strong computational scalability, allowing it to enhance exploration efficiency while maintaining controlled computational costs.
> >
> > [1] Chen, Weize, et al. "Agentverse: Facilitating multi-agent collaboration and exploring emergent behaviors in agents." arXiv preprint arXiv:2308.10848 2.4 (2023): 6.
> >
> > [2] Zhang, Bin, et al. "Controlling large language model-based agents for large-scale decision-making: An actor-critic approach." arXiv preprint arXiv:2311.13884 (2023).

---

### Review · Reviewer_B3N4 · 2025-03-08

**Summary Of Contributions:**

The authors propose a novel framework, ​eSpark, to automate action space pruning using Large Language Models, aiming to enhance exploration efficiency and policy performance in MARL.

**Audience:**

Yes

**Broader Impact Concerns:**

Not applicable.

**Claims And Evidence:**

Yes

**Requested Changes:**

Please refer to the above weakness part.

**Strengths And Weaknesses:**

**Strength:**
1.  Introducing LLMs to MARL for action space pruning.
2. Extensive experiments and good ablation studies.

**Weakness:**
1. The effectiveness of the eSpark framework relies heavily on the quality of LLM outputs. Errors in exploration function generation or feedback handling could degrade performance. Could the authors provide justification for this reliance?

2. The complexity of the proposed algorithm is not explicitly discussed. A detailed analysis would help assess its feasibility in practical applications.

3. The paper does not compare eSpark with other LLM-based approaches, making it difficult to evaluate its relative advantages. A comparative analysis would strengthen the contribution.

4. Are there alternative choices for prompts? Exploring different prompt designs could provide insights into the framework's robustness.

5. While tested on industrial tasks, broader applicability (e.g., robotics) needs exploration.

6. The presentation is generally good and easy to follow, but there are some places that can be improved. Just to name a few examples: is that correct to denote policy \pi as \pi: O\times A\rightarrow [0,1]? What is the purpose of defining exploration function E? What is the significance of proposition 1?

---

> ### Author Response · Authors · 2025-03-18
>
> We thank the reviewer for the sincere comments!  We hope we can address your concerns below.
>
> **Q1**: The effectiveness of the eSpark framework relies heavily on the quality of LLM outputs. Errors in exploration function generation or feedback handling could degrade performance. Could the authors provide justification for this reliance?
>
> **A1**: eSpark relies on the LLM generator to create and refine exploration functions. Therefore, minimizing errors or suboptimal outputs during generation is crucial for maintaining eSpark’s performance. We have implemented two methods to address errors or suboptimality in the generated exploration functions. These issues primarily stem from two sources:
>
> 1. Logical errors: The generated exploration function may contain flaws such as failing to account for task transitions or misusing variables.
> 2. Design flaws and hyperparameter selection: The function may have inadequately designed mask structures or suboptimal hyperparameters.
>
> To tackle logical errors, we employ a dedicated **LLM checker** that verifies each generated exploration function for logical inconsistencies, as detailed in Section 4.1. This role-assignment approach has proven effective in previous studies [7,8].
>
> For design flaws and suboptimal hyperparameters, we utilize **evolutionary search** to iteratively identify and select the best-performing exploration functions, as described in Section 4.2. This process naturally eliminates functions with inadequate designs or poor performance, excluding them from subsequent iterations.
>
> By combining these two mechanisms, we effectively mitigate the risk of performance degradation by either rectifying or excluding errors and suboptimal outputs from the LLM.
>
> **Q2**: The complexity of the proposed algorithm is not explicitly discussed. A detailed analysis would help assess its feasibility in practical applications.
>
> **A2**: We agree that complexity analysis is essential for assessing eSpark’s feasibility in large-scale applications. We provide a detailed time complexity analysis of eSpark, including:
>
> ​	1.	**Exploration function generation** (computational cost of LLM inference), primarily determined by the time required for the LLM to generate code.
>
> ​	2.	**Evolutionary search** (computational complexity of searching for the optimal strategy), which involves parallel evaluation of multiple strategies.
>
> ​	3.	**Policy feedback and refinement** (computational overhead for LLM processing and adjusting exploration functions).
>
> Since the generation, evolutionary search, and feedback processes of different exploration functions are independent of each other, we have implemented a highly parallelized code framework in our repository. This ensures that the computational cost does not increase linearly with the number of exploration functions. We present the runtime costs of eSpark compared to several MARL baselines across different scenarios in Table 17 in Appendix F.1.
>
> **Q3**: The paper does not compare eSpark with other LLM-based approaches, making it difficult to evaluate its relative advantages. A comparative analysis would strengthen the contribution.
>
> **A3**: We agree with the reviewer’s perspective on the importance of comparing eSpark with existing LLM-based methods. However, we find it challenging to adapt existing LLM-based methods to our task. Most existing LLM-based systems focus on textual-related tasks, and our task presents the following difficulties:
>
> 1. **No Abstract Actions**: The environments used in prior studies [1,2,4] generally include well-designed textual action interfaces or directly utilize text as actions. These abstract actions always consist of a series of low-level actions contain explicit meaning, reducing the cost of LLM calls while being easy to evaluate. In contrast, most data-driven RL environments provide **only** **single-step action interface**, making it difficult for an LLM to evaluate the consequences of each individual action effectively and increasing the calling cost.
> 2. **Multi-Agent Environments**: The environments considered in [1-4] are primarily single-agent settings. In such scenarios, LLM-based methods only need to evaluate actions and observations for a single agent, avoiding the complexity of joint action spaces. However, in MARL tasks, the action space and the number of actions to output grow significantly as the number of agents increases. Directly using LLM-based methods to assist in action selection for MARL tasks would result in prohibitive computational costs.
>
> To the best of our knowledge, we have not found any existing algorithms that can be directly applied to our task. We would greatly appreciate it if the reviewers could point us toward any existing methods that might address these challenges.

---

> > ### Author Response · Authors · 2025-03-18
> >
> > **Q4**: Are there alternative choices for prompts? Exploring different prompt designs could provide insights into the framework's robustness.
> >
> > Prompt engineering is crucial for tasks involving LLMs. During eSpark's development, we experimented with various ablations (e.g., excluding the LLM checker, modifying feedback instructions, adding or removing task-related components, shown in Section 5.4 and 5.5) and RL formulations (shown in Appendix G). These modifications influenced the LLM's generation, comprehension, and refinement abilities, thereby impacting the quality of exploration functions.
> >
> > When finalizing the prompt design, we adhered to two key principles:
> >
> > 1. Minimal manual input: Users need only provide essential environment-specific information. The framework automatically handles exploration function generation, evolutionary search, and function refinement through fixed prompts, ensuring quick adaptability across different environments.
> > 2. High-quality generation: The LLM must understand task requirements, conditions, and feedback to produce exploration functions that are executable, task-compliant, and effective in guiding MARL to explore redundant action spaces.
> >
> > In our final design, users input only a few RL formulations (mostly derivable from the environment) and a task description. Our framework then automatically generates exploration functions, trains models, and builds feedback loops. Experimental results show that our method consistently outperforms baselines across most scenarios.
> >
> > **Q5**: While tested on industrial tasks, broader applicability (e.g., robotics) needs exploration.
> >
> > **A5**: We appreciate the suggestion. Other domains like robotic tasks may involve **continuous action spaces**, whereas eSpark currently focuses on exploring and pruning in **discrete action spaces**. We agree that this is an interesting research direction. In future work, we plan to extend eSpark to support pruning in continuous action spaces, for example, by learning action sampling strategies based on Gaussian processes or variational autoencoders (VAEs), to evaluate eSpark’s applicability in a broader range of domains.
> >
> > **Q6**: The presentation is generally good and easy to follow, but there are some places that can be improved. Just to name a few examples: is that correct to denote policy \pi as \pi: O\times A\rightarrow [0,1]? What is the purpose of defining exploration function E? What is the significance of proposition 1?
> >
> > We will address the reviewer’s concerns from the following points:
> >
> > 1. Regarding the definition of policy **$\pi$**, an agent’s policy is typically represented as a mapping from the observation-action pair to a probability distribution, i.e., **$\pi: O\times A\rightarrow [0,1]$**. This representation is similar to the transition function definitions of [5,6], where $P:S\times A\times S\rightarrow[0,1]$.
> >
> >
> >
> > 2. The exploration function $E$ is the mathematical definition of the action mask generated by the LLM, serving as an essential foundation for the formal proof of Proposition 1.
> >
> >
> >
> > 3. Regarding Proposition 1, its core contributions include:
> >
> > •	The first point of **Proposition 1** proves that a **non-trivial exploration function $E$ can reduce the policy space size**, thereby improving exploration efficiency.
> >
> > •	The second point of **Proposition 1** proves that **for any policy $\pi_k$, there always exists some $E$ such that $\pi_k^E$ outperforms the original policy**.
> >
> > This theoretical proof demonstrates the effectiveness of the exploration function, supporting eSpark in reducing the agent’s search space through intelligent pruning while improving the original policy.
> >
> > [1] Zhuang, Yuchen, et al. "Toolchain*: Efficient Action Space Navigation in Large Language Models with A* Search." ICLR (2024).
> >
> > [2] Zhang, Shenao, et al. "How Can LLM Guide RL? A Value-Based Approach." arXiv preprint arXiv:2402.16181 (2024).
> >
> > [3] Wang, Xingyao, et al. "Executable Code Actions Elicit Better LLM Agents." ICML 2024.
> >
> > [4] Yan, Xue, et al. "Efficient Reinforcement Learning with Large Language Model Priors." arXiv preprint arXiv:2410.07927 (2024).
> >
> > [5] Zhong, Yifan, et al. "Heterogeneous-agent reinforcement learning." *Journal of Machine Learning Research* 25.32 (2024): 1-67.
> >
> > [6] Kuba, Jakub Grudzien, et al. "Trust region policy optimisation in multi-agent reinforcement learning." *arXiv preprint arXiv:2109.11251* (2021).
> >
> > [7] Chen, Weize, et al. "Agentverse: Facilitating multi-agent collaboration and exploring emergent behaviors in agents." arXiv preprint arXiv:2308.10848 (2023).
> >
> > [8] Zhang, Bin, et al. "Controlling large language model-based agents for large-scale decision-making: An actor-critic approach." arXiv preprint arXiv:2311.13884 (2023).

---

> > > ### Comment · Reviewer_B3N4 · 2025-04-18
> > >
> > > The reviewer thanks the authors for their detailed response. Most concerns have been addressed. One suggestion is for the authors to explore using different LLMs beyond GPT-4 to assess whether model choice leads to performance variations.

---

> > > > ### Author Response · Authors · 2025-05-08
> > > >
> > > > We thank the reviewer for the constructive feedback. In Appendix G, we have included a comparison of eSpark’s performance using different LLMs across two scenarios in the MABIM environment. We find that the performance of eSpark with GPT-4o is comparable to that with GPT-4. When using GPT-3.5, a slight performance drop is observed. Using DeepSeek-V3, although the performance is still inferior to GPT-4, it remains competitive, demonstrating that eSpark relies on the LLM’s capability for understanding and code generation to effectively generate exploration functions and improve policies.

---

### Comment · Reviewer_oyHM · 2025-02-26
**eSpark leverages LLMs to create exploration functions that enhance MARL algorithms, guiding exploration and learning through action space pruning. It bypasses complex prompt engineering and integrates with existing MARL algorithms.**

Strengths:
• Addresses action space explosion: The paper tackles the problem of combinatorial explosion of the action space in MARL, which leads to algorithmic instability and difficulty in convergence.
• eSpark, in a novel zero-shot approach, utilizes LLMs to generate exploration functions from task descriptions and environmental rules. This innovative method streamlines Multi-Agent Reinforcement Learning (MARL) by pruning unnecessary actions.
• Comprehensive methodology: The eSpark framework includes three components: zero-shot generation of exploration functions, evolutionary search for the best-performing MARL policy, and detailed feedback on the policy's performance to improve the exploration functions.
• Demonstrated effectiveness: eSpark consistently outperforms the combined MARL algorithm in inventory management and traffic light control tasks, achieving an average performance gain of 34.4% and 9.9% respectively. It also shows a 29.7% improvement in scalability challenges with over 500 agents.
• LLM checker and detailed reward feedback: The LLM checker reviews the LLM code generator's output to verify the accuracy of the code. This collaboration enhances the quality and efficacy of the generated content.

Areas for Improvement:
• Homogeneous agents: eSpark is currently only applicable to tasks involving homogeneous agents.
• Informative policy feedback: eSpark benefits from policy feedback to refine the exploration functions. When feedback is not informative regarding how to modify the exploration, eSpark may struggle to improve and need extra expert input for effective reflection.


Recommendation:

The paper's innovative framework for exploration in MARL, using LLMs to generate exploration functions, is a significant contribution. Its scalability, comprehensive methodology and proven effectiveness make it a valuable tool for MARL research.

Recommendation: Accept.

---

> ### Author Response · Authors · 2025-03-18
>
> Thank you for your detailed feedback and review! We are encouraged that you recognize the motivation of our approach in LLM based action space pruning and appreciate the promising empirical results of eSpark. Below we give the response to your concerns.
>
> **Q1**: Homogeneous agents: eSpark is currently only applicable to tasks involving homogeneous agents.
>
> **A1**: We acknowledge that this is a limitation of our current work, which we have already stated in Section 6. However, we have also proposed some potential solutions. For heterogeneous agent scenarios, we proposed to extend eSpark to by grouping agents based on their distinct action spaces and reward functions, then providing unique exploration functions for each group. This approach would enable eSpark to adapt to heterogeneous agent environments with minimal modifications.
>
> **Q2**: Informative policy feedback: eSpark benefits from policy feedback to refine the exploration functions. When feedback is not informative regarding how to modify the exploration, eSpark may struggle to improve and need extra expert input for effective reflection.
>
> **A2**: We agree that the quality of policy feedback is crucial for the effectiveness of eSpark. We have already discussed the role of policy feedback in Section 4.3 and validated the importance of detailed reward feedback through ablation experiments in Section 5.5. Additionally, we acknowledge that in some tasks, such as environments with sparse rewards, relying solely on environmental feedback may not be sufficient to automatically generate effective exploration functions. In future work, we plan to incorporate expert knowledge or self-supervised learning methods to enhance eSpark’s adaptability when feedback is insufficient.

---

### Author Response · Authors · 2025-03-18
**General Response**

We sincerely appreciate the insightful suggestions and valuable feedback provided by all reviewers. We're delighted that our reviewers recognize the motivation behind eSpark's innovative use of LLMs for action space pruning. We've addressed each individual concern raised by the reviewers in separate, detailed responses. To enhance the paper's clarity and readability, we have made the following revisions:

1.	We clarified some misunderstandings the reviewers had regarding our method and experiments.
2.	We reorganized the Related Work section to better distinguish between LLMs used for code generation and those used for RL/MARL.
3.	We added a summary table of different variable definitions in Appendix A.
4.	We consistently use subscripts for the k-th agent to enhance readability.

Thanks again.

---

### Decision · Action_Editor_qC3L · 2025-05-20

**Recommendation:** Accept as is

**Comment:**

Reviewers found the use of LLMs for action space pruning in MARL, its methodology, and demonstrated effectiveness as novel advancement to MARL. They confirmed the claims were supported and the paper relevant to the TMLR audience. Initial concerns included the reliance on LLM output quality and handling errors, the lack of detailed computational cost analysis of LLM usage, the need for comparison with other LLMs, exploring prompt sensitivity, applicability to heterogeneous agents and other domains like robotics, and the need for informative policy feedback, especially in sparse reward settings.

The revision addressed these points by explaining error handling mechanisms, providing detailed computational cost analysis and experiments with different LLMs (including open-source), clarifying notation, and discussing limitations and the importance of detailed feedback,

Reviewers "B3N4" and "GmtW" explicitly stated that their initial concerns were addressed by the authors' detailed responses and revisions, appreciating the added analysis and experiments with open-source LLMs and computational cost. All reviewers ultimately recommended acceptance for publication in TMLR. For example: oyHM: "The paper's innovative framework for exploration in MARL, using LLMs to generate exploration functions, is a significant contribution. Its scalability, comprehensive methodology and proven effectiveness make it a valuable tool for MARL research."

**Audience:**

The paper is relevant to MARL community. Reviewers noted that using LLMs for action space pruning in MARL is a "significant contribution" and a "valuable tool for MARL research". The potential for extending the method to "more general MA systems (in the context of LLM agent systems)" was also highlighted as a source of potential impact.

**Claims And Evidence:**

The paper introduces eSpark (Evolutionary action SPAce Reduction with Knowledge), to address action space explosion in Multi-Agent Reinforcement Learning (MARL). eSpark utilizes LLMs to generate exploration functions from basic task descriptions in a zero-shot manner, which are then used to prune redundant or irrelevant state-action pairs and guide the MARL exploration process. eSpark outperforms base MARL algorithms across 15 scenarios, achieving significant performance gains and demonstrating strong scalability to environments with a large number of agents.

Further support comes from ablation studies which show that key components like the LLM-based pruning, retention training, LLM checker, and detailed policy feedback are critical to eSpark's performance. Theoretical analysis also supports the idea that intelligent exploration functions can refine the policy space and enhance learning efficiency without losing the ability to find optimal policies.

Overall, the paper addresses a relevant topic, with the well-supported main claims that would be a welcome contribution to furthering MARL research.